# Distributed neural representation of saliency controlled value and category during anticipation of rewards and punishments

Zhihao Zhang [1,2,6], Jennifer Fanning[2], Daniel B. Ehrlich[1,2], Wenting Chen[2], Daeyeol Lee [1,3,4,5] & Ifat Levy [1,2,3,5]

An extensive literature from cognitive neuroscience examines the neural representation of value, but interpretations of these existing results are often complicated by the potential confound of saliency. At the same time, recent attempts to dissociate neural signals of value and saliency have not addressed their relationship with category information. Using a multi-category valuation task that incorporates rewards and punishments of different nature, we identify distributed neural representation of value, saliency, and category during outcome anticipation. Moreover, we reveal category encoding in multi-voxel blood-oxygen-level-dependent activity patterns of the vmPFC and the striatum that coexist with value signals. These results help clarify ambiguities regarding value and saliency encoding in the human brain and their category independence, lending strong support to the neural "common currency" hypothesis. Our results also point to potential novel mechanisms of integrating multiple aspects of decision-related information.

[1] Interdepartmental Neuroscience Program, Yale University, New Haven, CT 06520, USA. [2] Department of Comparative Medicine, Yale School of Medicine, New Haven, CT 06520, USA. [3] Department of Neuroscience, Yale School of Medicine, New Haven, CT 06510, USA. [4] Department of Psychiatry, Yale School of Medicine, New Haven, CT 06511, USA. [5] Department of Psychology, Yale University, New Haven, CT 06520, USA. [6] Haas School of Business, University of California, Berkeley, CA 94720, USA; Department of Neurology, University of California, San Francisco, CA 94158, USA. Correspondence and requests for materials should be addressed to I.L. (email: ifat.levy@yale.edu)

To make a decision, one needs to compare the subjective values of available options[1–3]. In many cases, these options are of fundamentally different nature, for example, an ice cream cone and a magazine. To facilitate such comparisons, the brain needs access to category-general, "common-currency", representations of subjective value[4].

Aiming to identify such representations, previous fMRI research has examined value encoding across different categories. Several studies identified overlapping value representations of items from different categories[5–11], a finding that is consistent with a pure category-general value representation, but is not sufficient to prove it. Stronger evidence for the neural common currency hypothesis came from two studies that used activation patterns in the ventromedial prefrontal cortex (vmPFC) to predict value or preferences across categories[12, 13,]. These studies,

however, were restricted to the domain of rewards, raising the possibility that these activation patterns encode saliency rather than value. While value quantifies how good or bad something is, saliency signals its significance. The saliency of available options is also an important variable in the choice process. An option of higher saliency elicits higher levels of attention and emotional arousal, as well as stronger orientation and motor preparation. Clearly, saliency and value are strongly correlated in the reward domain—a more rewarding outcome has higher value (it is more desirable), as well as higher saliency (as it is more important)[14]. Similarly, in the domain of punishments, saliency and value are negatively correlated. Identifying true value signals, which are distinct from saliency signals, therefore requires the inclusion of both reward and punishment categories in the experimental design. In such a design, moving from very negative to very

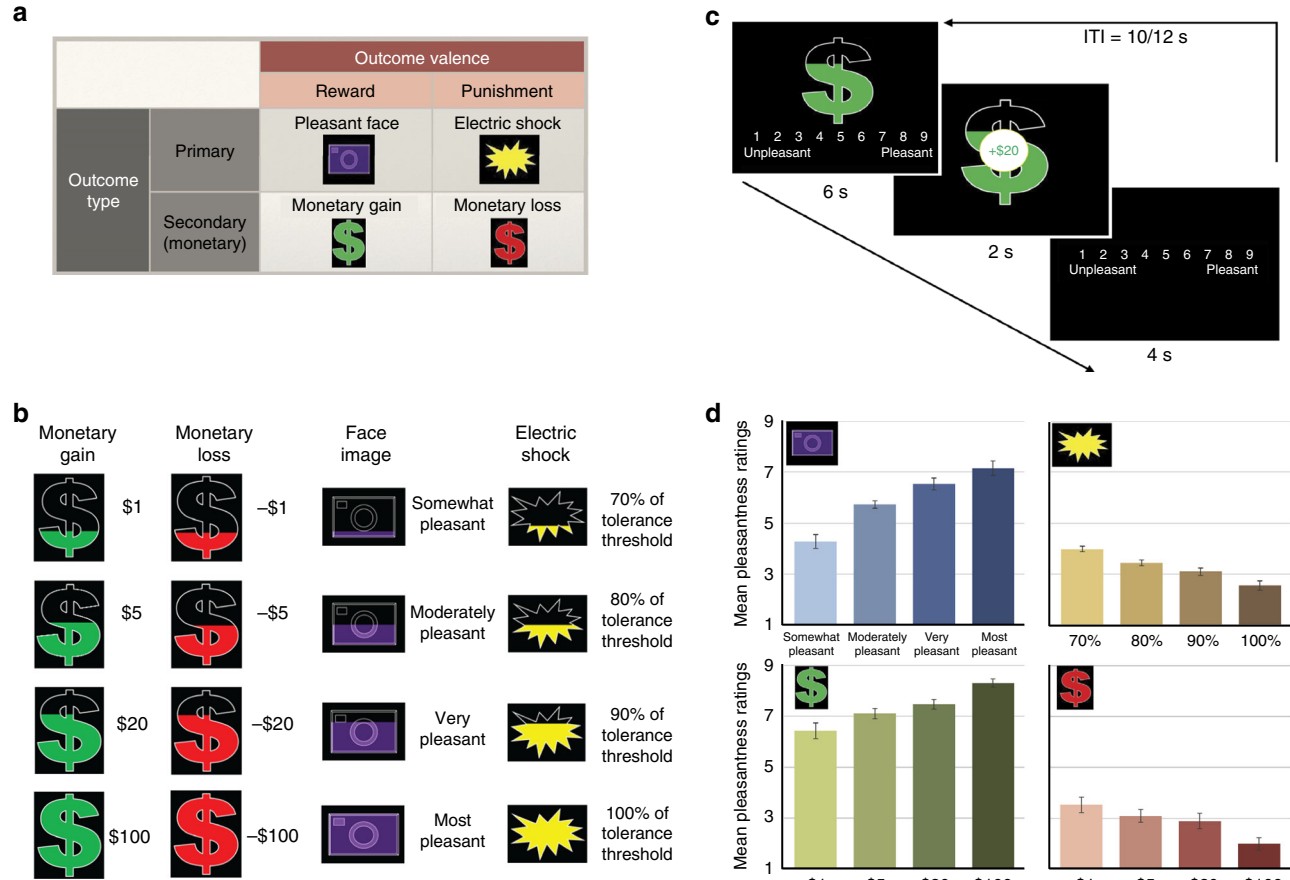

**Fig. 1** Multi-category valuation paradigm and behavioral pleasantness ratings. **a** Four outcome categories were included in the paradigm. These categories, resulting from the crossing between outcome valence (reward/punishment) and outcome type (primary/secondary), were: viewing pleasant faces, receiving electric shocks on one's ankle, monetary gains, and monetary losses. Each of these outcome categories was represented by a distinct shape, which the participants learned beforehand. These shapes served as cues that predicted the potential upcoming outcome in the current trial. **b** Four levels/intensities were included within each outcome category. Participants only received the predicted outcome in 1/3 of the trials; in the remaining trials nothing was delivered. In those actualized trials, participants could receive one of four different levels/intensities of the predicted outcome, which was indicated by the levels to which the corresponding shapes were filled. For instance, in monetary gains, the amount of money delivered could be $1, $5, $20, or $100. For other outcome categories see the Methods section. **c** Trial timeline. Each trial began with a cue presentation. Participants had to provide their pleasantness rating for the cue within 5.5 s. Two buttons on a botton box moved a cursor to the left and to the right along the 1–9 scale, and a third button was used to register the selection. A brief delay period of 0.5 s and a presentation of the outcome for 2 s then followed. For actualized shock trials, a 2 ms electric shock of the specified level was delivered during this time. For non-actualized trials of all categories, the message "No Outcome" appeared on the screen. Participants then had another 4 s to rate the outcome pleasantness. The trial length was held constant at 12 s, and the inter-trial interval was jittered between 10 s and 12 s. **d** Pleasantness ratings for cues predicting outcomes of different levels and of different categories (average across 18 participants, error bars denote SEM). A statistically significant two-way interaction between valence and magnitude indicated opposing trends of pleasantness ratings as a function of magnitude in reward and punishment categories. See Supplementary Fig. 2 for the pleasantness ratings for delivered and non-delivered outcomes

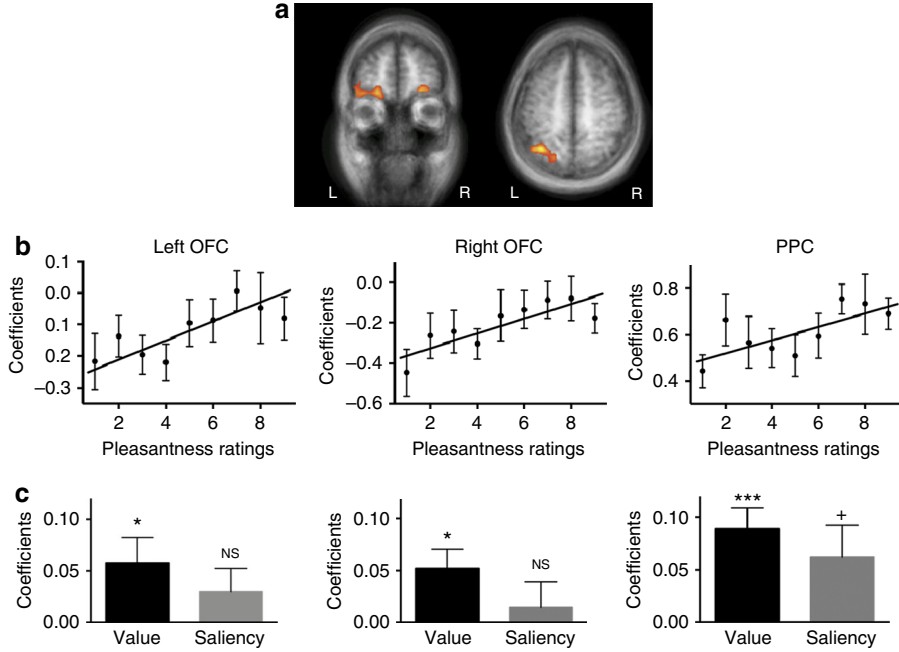

**Fig. 2** Univariate value signals in the brain. **a** BOLD activity of the bilateral OFC and the left PPC significantly correlated with cue value (per-voxel $p < 0.005$ and cluster-size thresholding at FWE $p < 0.05$). **b** Activity levels in these regions showed a positive slope on value (pleasantness ratings). Data were extracted from ROIs defined by a LOSO procedure ($n = 18$). **c** The effects of value and saliency on the activity of these brain regions were assessed in the same ROIs generated by the LOSO procedure, using a GLM with both value and saliency parametric regressors. Bilateral OFC demonstrated specificity to value, while PPC also showed a mild saliency effect. OFC orbitofrontal cortex. PPC posterior parietal cortex

positive stimuli generates a monotonically increasing profile for value and a U-shape function for saliency, allowing separation of their neural encoding. Although a few recent studies did examine the neural processing of both positive and negative outcomes[15, 16,], each of these studies used a single outcome type (e.g., monetary gains and losses). Whether the neural signals of value and saliency identified in these studies generalize across different categories remains unclear. Furthermore, the use of different analytic methodologies across previous studies also poses a nontrivial challenge for comparing and integrating their findings.

Our primary goal in this study was to identify the neural encoding of true value, controlling for saliency, across different outcome categories. To this end, a functional MRI paradigm, incorporating valuation of rewards and punishments of different nature, was necessary. Our experimental design also allowed us to explore the neural encoding of category information during valuation, as well as its interaction with the value signal. The representation of category identity has primarily been the subject of studies on semantic memory and conceptual knowledge, with little consideration of valuation[17–19]. Information about the nature and category of each option, however, is integral to making optimal decisions. Category information may be a key input to the computation of value, guided by the current motivational goals. For example, an ice cream and a magazine may both be valuable, but depending on your hunger level, you may prefer one or the other. We therefore hypothesized that category information may coexist with value signals to facilitate their flexible integration.

To the best of our knowledge, despite the interconnected relationships among these three variables—value, saliency, and category—no study so far has examined their neural representations simultaneously within the same context. In our fMRI paradigm, participants evaluated cues that predicted the probabilistic delivery of positive and negative outcomes, of various types and intensities. Our results reveal neural patterns for value,

which are distinct from both perceptual and value-based saliency signals, as well as distributed category-information encoding. Importantly, we found that activation patterns in both the vmPFC and the ventral striatum, areas heavily implicated in value representation, also provided category information. These findings help clarify existing ambiguities regarding "common-currency" value encoding in the human brain, and provide new insights on how category information may be integrated with value signals.

## Results

**Pleasantness ratings for cues**. Outcome categories included both rewards and punishments, of both primary (pleasant face images and electric shocks) and secondary (monetary gains and losses) nature (Fig. 1a). Four magnitude or intensity levels were used for each outcome type (Fig. 1b), so that participants experienced wide ranges of value and saliency. Pleasantness ratings (Fig. 1c), rather than choice behavior, were used as a measure of subjective value to allow examination of value signals that are not contaminated by choice and comparator signals. Mean pleasantness ratings for cues predicting different magnitudes of outcomes from different categories are presented in Fig. 1d. In the reward categories (monetary gains and pleasant faces), these pleasantness ratings increased as the level or magnitude of predicted outcomes increased. Conversely, in the punishment categories (monetary losses and electric shocks), ratings decreased as a function of the magnitude of predicted outcomes.

A three-way repeated measures ANOVA was used to determine the effect of outcome valence (reward/punishment), modality (primary/secondary), and magnitude on pleasantness ratings across participants. The two-way interaction between valence and magnitude was statistically significant ($F_{(2.219, 37.726)} = 91.877$, $p < 1e{-}6$), indicating opposing trends of pleasantness ratings as a function of magnitude in reward and punishment

**Table 1 Brain regions that showed univariate value or saliency responses (outcome expectancy) during the cue period**

| Contrast | Region | Side | Mean t statistic | Peak Talairach coordinates | | | Cluster size (number of voxels) |
|---|---|---|---|---|---|---|---|
| | | | | x | y | z | |
| Value | Posterior parietal cortex | L | 3.96 | −35 | −60 | 48 | 127 |
| | Lateral orbitofrontal cortex | R | 3.70 | 34 | 36 | 2 | 101 |
| | Lateral orbitofrontal cortex | L | 3.57 | −25 | 47 | −5 | 71 |
| | Lingual gyrus | R | 4.24 | 18 | −80 | 4 | 825 |
| Saliency | Rostral anterior cingulate cortex | L/R | 3.59 | −15 | 38 | 15 | 496 |
| | Precentral gyrus | L | 3.65 | −36 | −23 | 48 | 315 |
| | Caudate/Striatum | L | 3.87 | −9 | 4 | 8 | 106 |
| | Caudate/Striatum | R | 3.41 | 10 | 6 | 8 | 79 |
| | Anterior insula | R | 3.67 | 36 | 13 | 3 | 219 |
| | Anterior insula | L | 3.75 | −37 | 17 | 8 | 100 |
| | Lingual gyrus | L/R | 3.94 | −10 | −61 | 5 | 1307 |

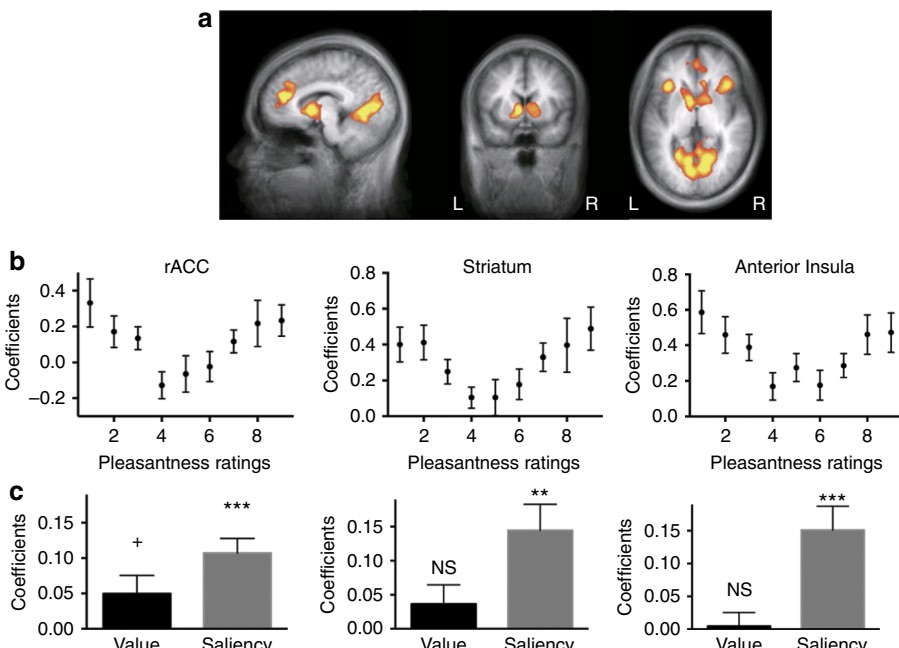

**Fig. 3** Univariate saliency signals in the brain. **a** BOLD activity in the rACC, the bilateral striatum and anterior insula significantly correlated with cue saliency (per-voxel $p < 0.005$ and cluster-size thresholding at FWE $p < 0.05$). **b** Activity levels of these regions showed a U-shaped profile as a function of pleasantness ratings. Data were extracted from ROIs defined by a LOSO procedure ($n = 18$). **c** The effects of value and saliency on the activity of these brain regions were assessed in the same LOSO ROIs, using a GLM with both value and saliency parametric regressors. Across all these regions, there was a significant saliency effect while value was not significant. rACC rostral anterior cingulate cortex

categories, consistent with the observations from Fig. 1d. There was no statistically significant three-way interaction between valence, modality, and magnitude ($F_{(1.415, 24.059)} = 0.908$, $p = 0.38$), so the valence × magnitude interaction was consistent in both primary and secondary modalities. There was also a statistically significant two-way valence × modality interaction ($F_{(1,17)} = 28.11$, $p = 6e−5$). The two-way modality × magnitude interaction was not statistically significant ($F_{(2.219, 37.726)} = 1.601$, $p = 0.21$).

**Whole-brain univariate analysis of value encoding.** One major goal in this study was to identify the encoding of value signals in the expectation phase, after controlling for saliency and regardless of outcome category. To this end, we constructed trial-by-trial cue value and saliency estimates based on each participant's own pleasantness ratings. Value was defined as the pleasantness rating in the cue period of the current trial; saliency was computed by taking the squared difference of the pleasantness rating and the neutral point 5 (see Supplementary Note 1 and Supplementary Fig. 1 for the rationale of this approach and potential caveats). We regressed BOLD signals against these trial-wise estimates in a general linear model to compute the contribution of value and saliency signals to the activity of each voxel. We then searched for voxels whose activity was positively correlated with value across participants (Fig. 2a, Table 1; for details, see Methods section). This analysis revealed univariate value signals in the left posterior parietal cortex (PPC; family-wise error rate (FWE) $p < 0.05$ cluster-size thresholding, threshold = 69 voxels, same below; center Talairach coordinates $X = −39$, $Y = −61$, $Z = 55$) and bilateral orbitofrontal cortex (OFC; $X = −21$, $Y = 50$, $Z = −5$ in the left hemisphere, $X = 30$, $Y = 29$, $Z = −2$ in the right hemisphere). To

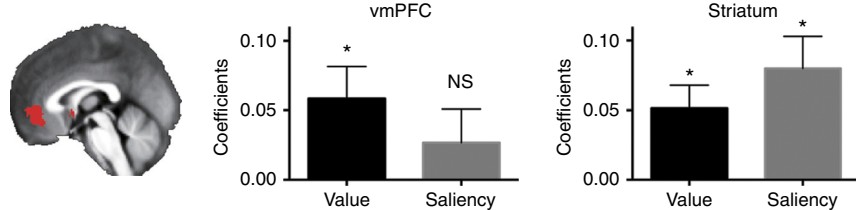

**Fig. 4** Region-of-interest univariate analysis. Univariate value and saliency signals in the vmPFC and the ventral striatum, defined based on a recent meta-analysis of the valuation system[20]. In the vmPFC, there was a significant effect of value, but not saliency, while in the ventral striatum, the effects of both value and saliency were statistically significant ($n = 18$). Error bars denote SEM. NS not significant; $^+p < 0.1$; $^*p < 0.05$; $^{***}p < 0.001$. vmPFC ventromedial prefrontal cortex

visualize the effect of value on activation in these regions, we extracted parameter estimates, determined by the leave-one-subject-out (LOSO) procedure, and plotted them as a function of pleasantness ratings for the cues (Fig. 2b). A mixed effects linear regression of parameter estimates vs. pleasantness rating showed significantly positive slopes for all three regions ($t_{(17)} = 3.82$, $p = 0.0014$ for PPC; $t_{(17)} = 2.45$, $p = 0.037$ for left OFC; $t_{(17)} = 4.23$, $p = 0.00056$ for right OFC). Using the same set of LOSO regions-of-interest (ROIs), we also directly quantified the strength of value and saliency effects on the BOLD activity in these brain regions (Fig. 2c). In both the left and the right OFC, there was a statistically significant effect of value ($t_{(17)} = 2.35$, $p = 0.031$ for left OFC; $t_{(17)} = 2.76$, $p = 0.014$ for right OFC), while saliency did not reach significance ($t_{(17)} = 1.39$, $p = 0.18$ for left OFC; $t_{(17)} = 0.58$, $p = 0.57$ for right OFC). In PPC, a strong value effect was found ($t_{(17)} = 4.27$, $p = 5.8e{-}4$), while the saliency predictor was marginally significant ($t_{(17)} = 1.97$, $p = 0.066$).

**Whole-brain univariate analysis of saliency encoding**. Our experimental paradigm also allowed us to search for brain areas whose activity is positively correlated with saliency after controlling for value. This analysis revealed a different set of brain regions (Fig. 3a and Table 1), including the rostral anterior cingulate cortex (rACC; FWE $p < 0.05$ cluster-size thresholding, threshold = 72 voxels, same below; center Talairach coordinates $X = -12$, $Y = 38$, $Z = 16$), bilateral striatum ($X = -9$, $Y = 5$, $Z = 4$ in the left hemisphere, $X = 9$, $Y = 8$, $Z = 4$ in the right hemisphere), left precentral gyrus ($X = -36$, $Y = -22$, $Z = 52$), bilateral anterior insula (AI; $X = -33$, $Y = 20$, $Z = 4$ in the left hemisphere, $X = 33$, $Y = 17$, $Z = -2$ in the right hemisphere), and visual areas. Similarly, plotting activation levels as a function of ratings in corresponding ROIs defined by the LOSO procedure clearly showed U-shaped patterns, consistent with predictions of saliency encoding (Fig. 3b). The activities of these ROIs also demonstrated statistically significant saliency effects (Fig. 3c; $t_{(17)} = 5.01$, $p = 0.00013$ for rACC; $t_{(17)} = 3.66$, $p = 0.0021$ for striatum; $t_{(17)} = 5.14$, $p = 0.0001$ for AI), but not value ($t_{(17)} = 1.26$, $p = 0.23$ for striatum; $t_{(17)} = 1.18$, $p = 0.26$ for AI), except for a marginally significant effect in the rACC ($t_{(17)} = 1.84$, $p = 0.084$). A control analysis verified that these findings were not likely to be driven by hand movement or the time it took participants to register their responses (see Supplementary Note 2 and Supplementary Methods for details). Furthermore, as our operationalization of saliency likely encompassed both value-based saliency and perceptual (visual) saliency, we conducted further analysis using separate estimates for these two types of saliency (Supplementary Note 3). This analysis revealed that value-based saliency and visual saliency mapped onto spatially distinct regions in the brain, with the former reflected in activation in rACC, striatum and anterior insula, and the latter reflected mostly in activation of sensory areas (Supplementary Fig. 4, and Supplementary Table 1).

**ROI analysis of univariate value and saliency signals**. In addition to the whole-brain analysis, we also examined the effects of value and of saliency on BOLD activity in a set of ROIs from a recent meta-analysis focusing on the valuation system[20], including the vmPFC and the ventral striatum (VS). vmPFC activity was significantly modulated by value ($t_{(17)} = 2.52$, $p = 0.022$), but not saliency ($t_{(17)} = 1.10$, $p = 0.29$), while in the VS, both value and saliency contributed significantly to the activation ($t_{(17)} = 3.11$, $p = 0.0064$ for value and $t_{(17)} = 3.48$, $p = 0.0029$ for saliency) (Fig. 4). This mixture of value and saliency representations in the VS is consistent with the findings of a previous study[15]. In addition, a small area in the VS was also identified in a whole-brain conjunction analysis of correlation with cue value and with cue saliency (center Talairach coordinates $X = -6$, $Y = 2$, $Z = -2$), if a less stringent statistical threshold were used (per-voxel $p < 0.005$ uncorrected, cluster size > 20 voxels).

**Univariate and multi-voxel encoding of category information**. Next, we examined univariate representations of category identity using a general linear model (GLM) with binary predictors for each of the four categories, and controlling for value and saliency (see Supplementary Note 4). While we did observe univariate category-related signals in several brain areas (Supplementary Tables 2 and 3, and Supplementary Fig. 5), value-related areas did not encode category identity in the univariate fashion. Information about outcome category (and other key variables) could still be encoded in multi-voxel activation patterns, rather than overall activation magnitude, in the same areas. To test that hypothesis, we used whole-brain searchlight representational similarity analysis (representational similarity analysis (RSA); Fig. 5a; see Methods for details). A 27-voxel ($3 \times 3 \times 3$) cubic searchlight was formed around each voxel, and a neural representational dissimilarity matrix (neural RDM) was constructed for each cube, computing the difference (dissimilarity) between voxel-wise patterns of neural activities in different conditions. We then compared the neural RDMs to predicted dissimilarities in neural patterns between conditions based on models of category identity, value, or saliency encoding. Figure 5b through 5d and Table 2 present areas of the brain in which activation patterns represented category, value, or saliency, respectively (per-voxel $p < 0.005$, FWE $p < 0.05$ cluster-size thresholds = 38, 25, 41 voxels, respectively). Importantly, Pearson correlational distance is insensitive to general changes of mean activation levels of the ROI, and thereby provides complementary information to univariate GLM analyses. This analysis revealed widespread, distributed multivariate representation of category information in many cortical areas, as well as several subcortical structures (Fig. 5b). Notably, these areas overlap substantially with the traditional univariate value areas, including the vmPFC and VS. Conversely, multi-voxel representations of value and saliency were more localized (Fig. 5c, d). Interestingly, both value and

saliency signals were encoded by multi-voxel patterns in adjacent, but potentially distinct, regions of PPC.

To account for the possibility that category effects may be driven by other category attributes, rather than its identity, we examined several additional model RDMs (Fig. 6; see Methods for details). These models tested the effects of the mean value of each category (Fig. 6a), whether the category is primary or secondary (modality; Fig. 6b), and the category valence (Fig. 6c), as well as the effect of potential interactions between valence and modality (Fig. 6d, e and Supplementary Table 4). These control models further confirmed the specific encoding of category identity. Very few brain areas showed significant multi-voxel encoding of the

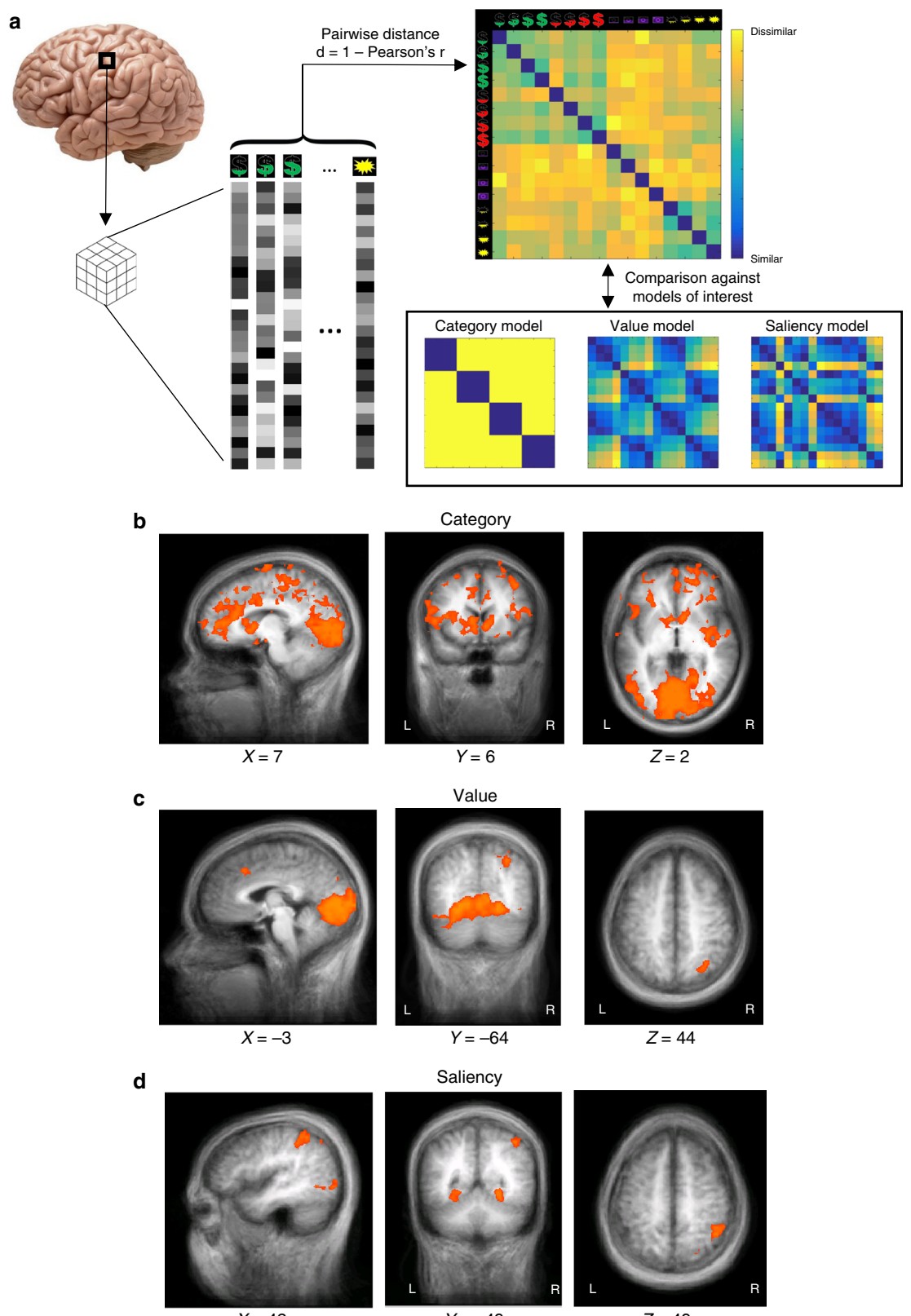

**Table 2 Brain regions with statistically significant multivariate encoding of category, value, or saliency as identified by representational similarity analysis**

| Candidate model | Region | Side | Mean z statistic | Peak Talairach coordinates | | | Cluster size (number of voxels) |
|---|---|---|---|---|---|---|---|
| | | | | x | y | z | |
| Category | Cuneus | L/R | 3.37 | 6 | −79 | 13 | 3422 |
| | Superior frontal gyrus | L | 3.30 | −31 | 48 | −22 | 248 |
| | Precuneus | R | 3.29 | −19 | −69 | 41 | 727 |
| | Caudate/Striatum | R | 3.28 | 19 | −75 | 9 | 106 |
| | Inferior parietal lobule | R | 3.32 | 40 | −77 | −1 | 70 |
| | Inferior frontal gyrus | R | 3.30 | 49 | 24 | 21 | 249 |
| | Middle frontal gyrus | R | 3.28 | 28 | 37 | −17 | 116 |
| | Anterior cingulate cortex | R | 3.30 | 22 | 45 | 4 | 447 |
| | Paracentral lobule | L | 3.26 | −3 | −33 | 56 | 60 |
| | Caudate/Striatum | R | 3.31 | 13 | 3 | 13 | 68 |
| | Cingulate gyrus | R | 3.26 | 10 | −2 | 29 | 63 |
| Value | Lingual gyrus | L/R | 3.16 | −7 | −81 | −3 | 2169 |
| | Inferior parietal lobule | R | 2.86 | 31 | −64 | 40 | 53 |
| | Medial frontal gyrus | L/R | 2.81 | 0 | 29 | 35 | 25 |
| Saliency | Inferior parietal lobule | L/R | 3.14 | 42 | −52 | 43 | 87 |
| | Middle occipital gyrus | R | 3.18 | 39 | −80 | 6 | 1876 |
| | Inferior temporal gyrus | L | 3.11 | −45 | −67 | 0 | 48 |

mean value of a category (Fig. 6a; per-voxel $p < 0.005$, FWE $p < 0.05$ cluster-size thresholding, threshold = 40 voxels). Moreover, modality or valence did not explain much of the variability in multi-voxel patterns across the brain: compared with the category model, the primary-secondary model and the positive-negative model were reflected in more circumscribed multi-voxel representations in the brain (Fig. 6b, c; per-voxel $p < 0.005$, FWE $p < 0.05$ cluster-size thresholding), and only had minor overlap with the regions that showed significant responses to the category model. Similarly, results from the models targeting potential interactions between valence and modality (Fig. 6d, e; per-voxel $p < 0.005$, FWE $p < 0.05$ cluster-size thresholding) revealed little overlap with the widespread reflection of category information observed in Fig. 5b.

We also performed RSA with four key candidate models in the ROIs identified by the univariate analysis (Supplementary Note 5, Supplementary Methods, and Supplementary Fig. 6). The category-identity model contributed significantly to the activation patterns in all of these ROIs, confirming the widespread multi-voxel coding of category information. The fact that brain areas that encode value or saliency in a univariate manner showed little multivariate representations of these quantities also demonstrates the independence of the GLM and RSA approaches.

**Multidimensional category and value signals in vmPFC and VS**. RSA revealed considerable spatial overlap between regions that showed multi-voxel encoding of category information and the well-established valuation areas, including the vmPFC and VS. This suggests an intriguing possibility: category information

and value may coexist within the same brain region, but in different forms. To probe this possibility we performed a data-driven analysis using principal component analysis (PCA), a methodology that reveals the internal structure in multi-dimensional data and does not assume particular forms of the signal. We expected that PCA would uncover both value and category information from the valuation areas, and that the loading structures of the identified neural signals may reveal distinct forms of encoding, consistent with the univariate and RSA findings.

Similar to RSA, we used the voxel-wise beta coefficients for the 16 cue conditions in all functional voxels of the vmPFC ROI as the input to PCA. The percentages of total variance explained by individual principal components (PCs) were consistent across participants (Fig. 7a). As there was no substantial improvement in total variance explained beyond the third PC, only the first 3 PCs were kept for further analysis. Across participants, these PCs accounted for $81.7 \pm 1.05\%$ (mean ± SEM) of the total variance. To demonstrate the relative contributions of the voxels in the vmPFC ROI to these PCs, we examined the histograms of voxel-wise loading coefficients, or the contributions of different voxels to each PC, for each participant and present the average distributions of the loading coefficients in Fig. 7b. The loading coefficients for the first PC were distributed along a narrow range of positive values consistently across participants, indicating that this PC resembles the mean activity of all voxels in this ROI. In contrast, the second and the third PCs showed more complicated loading structures with loading coefficients spanning both positive and negative values. This potentially represents

**Fig. 5** Whole-brain searchlight representational similarity analysis (RSA) of multi-voxel category, value, and saliency signals. **a** Schematic illustration of the procedure of the whole-brain searchlight RSA. In this analysis, activation patterns in the immediate neighborhood of every voxel (a 27-voxel cube with that voxel as the center) was examined. The construction of searchlight neural representational dissimilarity matrices (neural RDMs) from multi-voxel activity patterns in response to each condition in the paradigm was based on Pearson correlational distance. The neural RDMs were then compared with candidate model RDMs based on different hypotheses. Example theoretical candidate similarity matrices based on category identity, value, and saliency are shown. The latter two candidate matrices depended on pleasantness ratings and, therefore, differed between participants. The candidate matrices from one representative participant are shown here. **b–d** Whole-brain maps of the z-statistic of the resemblance between the neural RDMs and each of the candidate RDMs, including category **b**, value **c**, and saliency **d**. Statistical thresholds for all maps were per-voxel $p < 0.005$ and cluster-size thresholding at FWE $p < 0.05$. The human brain image in **a**, entitled "human brain on white background", by _DJ_ is sourced from https://www.flickr.com/photos/flamephoenix1991/8376271918 and licensed under CC BY 2.0 (https://creativecommons.org/licenses/by-sa/2.0/)

population coding by local activity patterns, instead of mean activation levels.

To examine whether these three PCs correspond to any of our variables of interest, including value, saliency, and category, we performed mixed-effects linear regressions of each PC on these variables (Fig. 7c). The first PC was significantly correlated with value ($t_{(16)} = 2.44$, $p = 0.027$), but not with saliency ($t_{(16)} = 0.98$, $p = 0.34$) or category (coded as three dummy variables, using the electric shock category as the reference category; all $p > 0.08$), and the difference between value and saliency effects was significant ($p = 0.012$). The second PC was not significantly correlated with any of the variables (all $p > 0.38$), while the third PC was significantly correlated with both value ($t_{(16)} = 3.40$, $p = 0.0036$) and category (all $p < 0.0231$), but not with saliency ($t_{(16)} = -0.138$, $p = 0.89$).

A parallel analysis on the VS ROI from the univariate analysis revealed similar, albeit slightly weaker, results (Supplementary Fig. 7). Across participants, the first three PCs in this ROI collectively accounted for $83.1 \pm 1.13\%$ (mean ± SEM) of the total variance. Loading structures were similar across participants: the first PC was analogous to the mean activity of all voxels, while the next two PCs had loading profiles consistent with multi-voxel patterns. Follow-up mixed-effects regressions showed that the first PC was significantly correlated with value ($t_{(16)} = 2.425$, $p = 0.028$), but not with saliency ($t_{(16)} = 1.07$, $p = 0.30$), or category (all $p > 0.27$). No significant effects were found for PC2, while PC3 showed trends towards significance for value ($t_{(16)} = 1.805$, $p = 0.09$) and category (compared to electric shocks: monetary gains $t_{(16)} = 2.304$, $p = 0.035$; monetary losses $t_{(16)} = -0.064$, $p = 0.95$; pleasant faces $t_{(16)} = 1.801$, $p = 0.09$). Taken together, these results demonstrate the multiplexing of univariate and multi-dimensional value and category signals in both vmPFC and VS. Results of PCA in other ROIs are reported in Supplementary Note 6.

## Discussion

In this study, we used a multi-category valuation task to identify pure category-general value encoding on a common scale, after accounting for saliency effects. We also explored the representation of category identity, a crucial input for valuation, and how it may integrate with value encoding. Our univariate analysis revealed regions whose activation magnitudes encode value, even after controlling for saliency, as well as regions in which activation scales with saliency after controlling for value. Multi-voxel pattern analysis, including RSA and PCA, offers a complementary view of the data and suggests that category information is encoded in a distributed manner across multiple brain regions, overlapping substantially with previously known valuation areas including the vmPFC and VS.

These findings contribute to the interpretation of results from prior studies of value encoding, which have not controlled for saliency confounds. Our results confirm the univariate representation of value in several brain regions that have been repeatedly implicated in value encoding in various value-based decision-making tasks, including vmPFC, OFC, PPC, and striatum[21, 22,]. Such value signals also seemed to be independent of the category identity of the evaluated outcome. It is also worth noting that although no choice was required in our paradigm, the localization of univariate representations in our study was largely consistent with many previous studies involving active decisions. Indeed, the notion of valuation processes that are independent of choice has received empirical support[23–28] (also see Supplementary Discussion of implications of different techniques for value elicitation). The present study extends these results by showing that this brain system for valuation in the absence of choice is capable of processing anticipation of rewards and punishments both on a continuous scale and across different categories.

The saliency areas we identified (rACC, striatum, and anterior insula) substantially overlap with the saliency regions identified in a prominent previous study that used appetitive and aversive foods to dissociate value from saliency signals[15]. The consistency of results across several different categories (monetary gains and losses, shocks, and foods) strongly supports the conclusion that saliency-like signals in these regions are category-independent. Furthermore, saliency as defined in this study may encompass both bottom-up perceptual saliency[29–31] and top-down value-based saliency[15, 16,]. In this study, both forms of saliency were examined, in order to facilitate the unambiguous identification of true value signals.

One caveat is the possibility that the saliency signals we observed resulted from a "valence-general" neural representation of value, rather than from a saliency encoding per se[32]. In such representation, neural activity scales with the absolute value of both rewards and punishments, resulting in a U-shaped activation profile as a function of value. In addition, given the resolution of fMRI, we cannot rule out the possibility that two distinct groups of neurons exist within the same voxel, one responding monotonically to increasing intensities of reward and the other to increasing punishments, thereby creating a saliency-like activity profile for the voxel. Further studies are needed to distinguish between these accounts.

Our study also examined the representation of category information. While we know that conceptual knowledge of category information is distributed in multiple brain areas[33, 34], very few studies examined categorization in the context of decision-making. One study showed both category-independent and category-specific value encoding in different sub-regions of the vmPFC/OFC using three categories of rewards[13]. Along the same lines, a recent study found identity-specific and identity-general encoding of reward value in OFC and vmPFC, respectively[35]. Finally, using repetition suppression, another study demonstrated encoding of reward identity in part of medial OFC[36]. To our knowledge, our study is the first to incorporate punishments to assess the effects of category on value processing. Using RSA, we found widespread representation of category identity in brain regions associated with univariate signals of value and saliency. By testing a series of control models, we ruled out the possibility that differences in the mean value, valence, or modality of different categories, were driving the significant category effects, rather than category identity itself. Importantly, these category-related activities occurred at the cue period, in which participants had not experienced any of the predicted outcomes, thus reflecting anticipation of specific categories, rather than actual experiences. It is also unlikely that sensory properties of the cues were the cause for these effects, because regions like vmPFC or striatum are unlikely to encode visual information strongly and passively.

Our findings that univariate value signals and multivariate category signals coexist in the vmPFC, and striatum provide a novel insight about the role of these areas in valuation. This multiplexing of two distinct signals in a single-brain region suggests an efficient coding strategy, which could increase the capacity of information encoded by the region and facilitate subsequent utilization of these two sources of information to guide behavior and monitor environmental feedback. This notion is supported by our PCA results, which identified a multi-dimensional value signal in the vmPFC, compatible with several previous studies[12, 13, 37, 38,]. The coexistence of value and category effects in the same principal component may explain why the whole-brain searchlight RSA did not identify this multi-voxel value code in the vmPFC. This finding bears particular

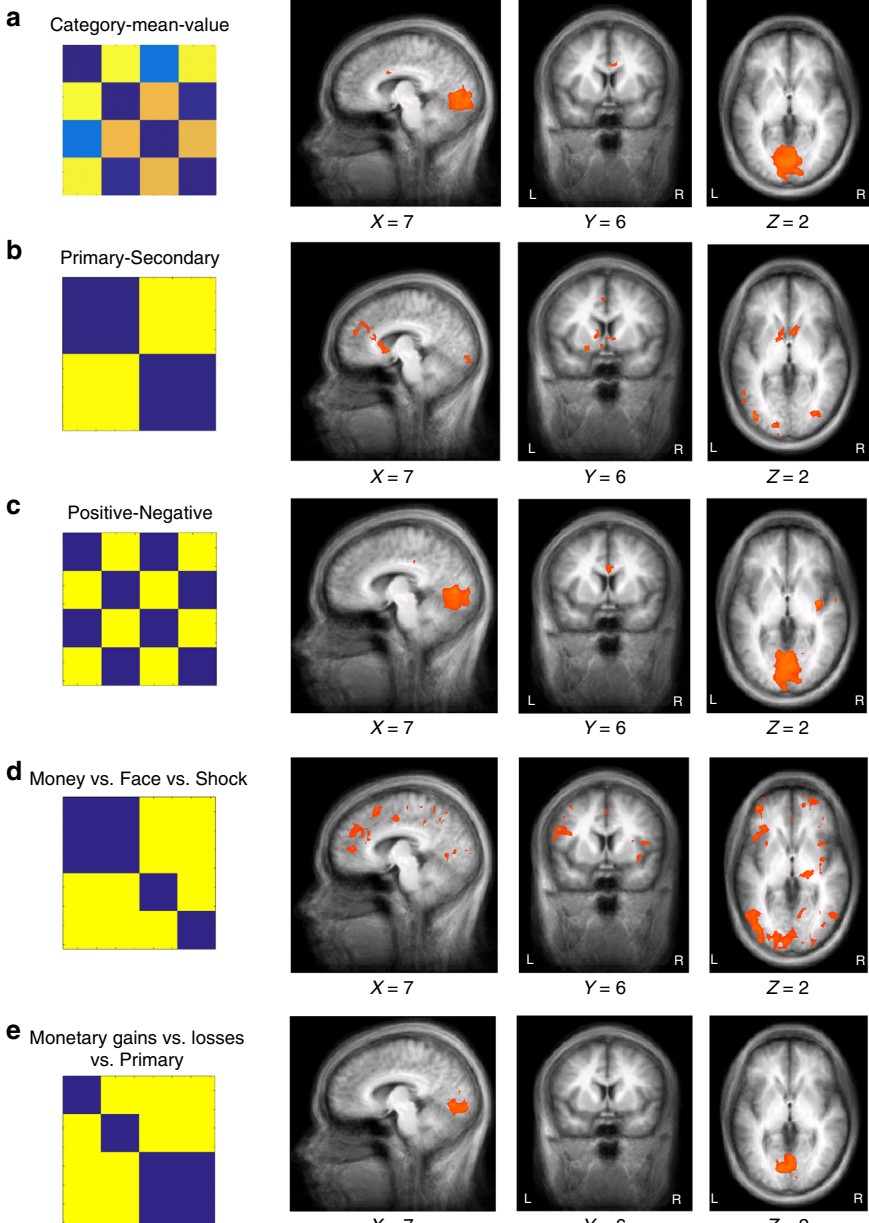

**Fig. 6** Whole-brain searchlight RSA for additional control models. Five additional control models for the category model are shown, together with their whole-brain searchlight statistical maps. In each panel, a pseudo colored candidate model RDM for the corresponding model is shown on the left. Warmer colors indicate greater distance and thereby larger dissimilarity. The order of the conditions is the same as in Fig. 5a. For each model, whole-brain maps of the z-statistic of the resemblance between the candidate model RDM and the neural RDMs are shown on the right. Statistical thresholds for all maps were per-voxel $p < 0.005$ and cluster-size thresholding at FWE $p < 0.05$ by simulation of 1000 random samples. To facilitate comparison, all maps are presented from the same planes as in Fig. 5b for the category model. **a** The category-mean-value model from a representative participant. **b** The primary-secondary model. Because this model does not involve ratings, all participants have the same model RDM (same below). **c** The positive-negative model. **d** The money vs. face vs. shock model. **e** The monetary gains vs. monetary losses vs. primary model

significance for attempts to decode subjective values and consumer preferences from neural data. The results are also of relevance for models of value computation. Prior research shows that vmPFC integrates multiple attribute-value signals from other, specialized, brain areas[39, 40,]. There is also evidence suggesting that in addition to comparisons of these integrated values, the choice process may also entail direct comparisons at the level of single attributes[41]. Moreover, recent research suggests that value may be determined, to a large extent, based on memories of individual instances of encountering the particular reward[42, 43,]. Current models of value and choice thus require the simultaneous

encoding of category-independent value and of category information (as well as other specific information of the evaluated option). Future research will need to determine how conceptual category knowledge in temporal regions[18] is transformed into the category information encoded in value areas. Such research will also need to reveal the nature of the multidimensional value signals in vmPFC (and perhaps also the VS), including their regional specificity and context dependence, temporal dynamics of the category information in value computation, and whether the effects we observed here extend beyond the four outcome categories included in our design.

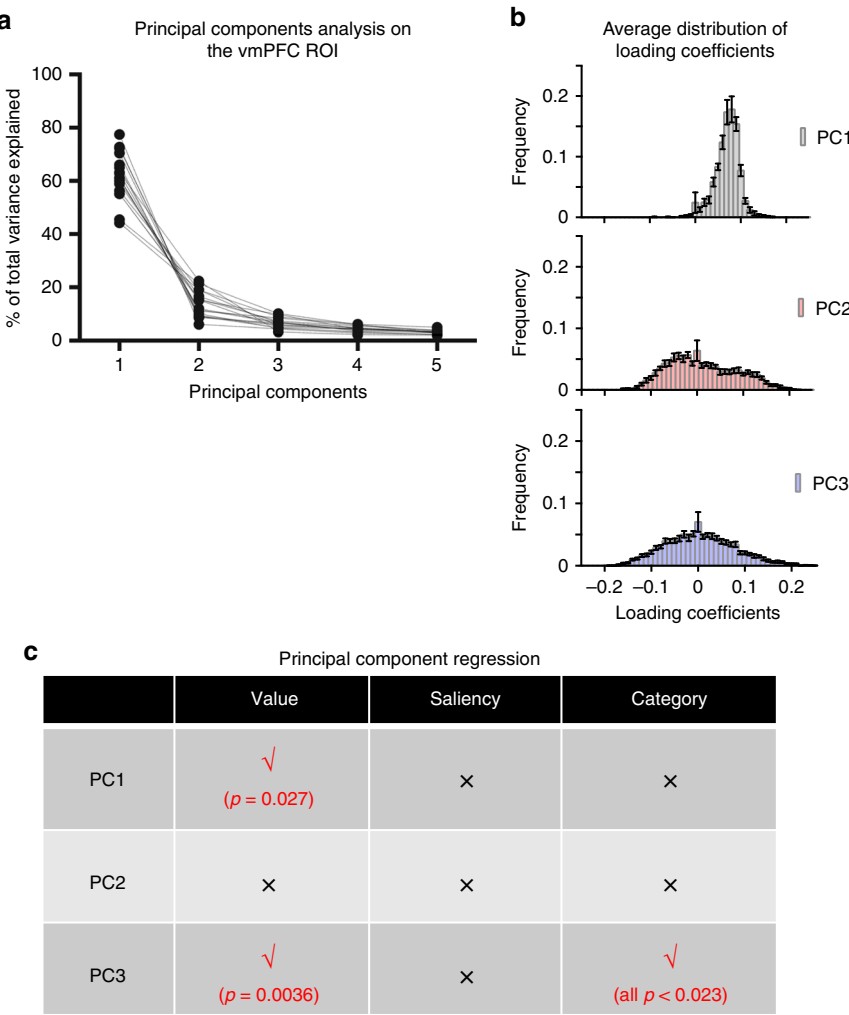

**Fig. 7** Principal component analysis of ensemble patterns in vmPFC. **a** Percentages of total variance explained by the first 5 PCs in each participant ($n = 17$). Each dot is a data point from a participant, and data from the same participant are connected by dashed lines. **b** Average histograms of loading coefficients for the first 3PCs. The frequency of loading coefficients falling under a particular bin was calculated separately for each participant. The mean and the SEM of frequencies in this bin were then obtained. This was repeated for all bins, spanning the entire range of loading coefficients. The mean frequencies in each bin are plotted here in the form of a histogram with error bars representing SEMs. **c** Summary of results of mixed-effect linear regression of the first 3 PCs on value, saliency, and category

This study also provides several methodological insights. The multivariate distance we used in RSA, the Pearson correlational distance, is insensitive to changes of overall activation magnitude, making the RSA orthogonal to the univariate GLM analysis[44]. This offers a unique advantage for the unambiguous identification of univariate and multidimensional coding of decision variables, which has proven difficult for other multi-voxel pattern analysis techniques, such as classification and support vector regression[16, 37,]. A recent study of value representation in the amygdala demonstrated the strength of this approach[45]. While mean activation was similar for different odors, differences in pleasantness ratings of these odors could account for (dis)similarity in activation patterns as measured by the Pearson distance. Similarly, univariate analysis and RSA with the Pearson distance uncovered multiplex encoding of arousal and affect in the medial OFC[46]. However, a more comprehensive whole-brain analysis of parallel univariate and multidimensional representations of value and related decision variables has yet to be reported.

Our study fills this gap in the literature. Consistent with the orthogonality of the two analytic techniques, there is little overlap between the univariate and multivariate representations (Figs. 2

and 5, Supplementary Fig. 6). Both methodologies identified brain regions that have been implicated in valuation under different contexts, including the OFC, PPC, and ventral striatum[16, 47, 48,]. They shed new light, however, on the particular nature of these signals in the brain. For instance, using support vector regression, both OFC and PPC were reported to encode predicted value of monetary outcomes[16]. Our results suggest that despite their apparent similarity, value codes in these two regions might rely on different implementations, as OFC shows exclusive univariate coding, while PPC seems to utilize both mean activity levels and multivoxel patterns.

One potential limitation of both univariate GLM and RSA approaches is the strong assumptions they make about the specific features of information coding by neural activities. This can be nicely overcome by PCA. As a widely used exploratory statistical technique in multivariate spaces, PCA makes very few, if any, assumptions about the nature of signals other than the constraint of linear combinations of individual units (voxels). PCA therefore offers a data-driven perspective on the functional organization of voxel-wise activities in ROI, which can be seen as an independent validation of both univariate and multivariate

analyses. Indeed, in all value or saliency ROIs, PCA invariably identified a mean-activity-like component that accounts for a majority of the total variance, and mixed-effects regressions of this component consistently confirmed results from the whole-brain and ROI GLM analysis. One notable exception is the lack of significant saliency effect in the first PC in the VS, even though the ROI GLM indicated both value and saliency encoding (Fig. 4). This could be due to the slight spread of loading coefficients for this PC, producing deviation from the true mean activity across all voxels. Regardless, noting that the PCA was based on a very different partition of conditions compared with the univariate GLM analysis (see Methods section) and that the mean-activity-like principal component evolved entirely from the covariance structure of the data itself, the overall consistency between PCA and univariate GLM results indicates the prominence of univariate encoding by magnitude of neural activity. Furthermore, PCA in the vmPFC and VS confirmed the presence of multi-voxel category signals as identified by the whole-brain searchlight RSA (Fig. 5), while also revealing more complicated ensemble encoding of multiplexing value and category signals (Fig. 7). In some other cases, though, results of RSA and PCA were complementary, for instance in rACC multi-voxel category encoding was implicated by RSA, while at least the first 3 PCs identified by PCA did not show effects of category. One reason for this might be that the RSA uncovered category signals which had more complicated structure than linear combinations of individual voxel activations. The use of multiple analytic techniques (univariate GLM, RSA, and PCA) in parallel is therefore beneficial for gaining a complete picture of the nature of neural representation of value, saliency, and category in these regions.

Understanding value encoding and its interaction with category is important because this is a fundamental cognitive function at almost every level of human behavior. Our results demonstrate value encoding when saliency is adequately controlled for, resolving ambiguities in the interpretation of prior results on value representation in the brain. Furthermore, our findings suggest that while a univariate neural common currency does exist, there is much richer information in the ensemble activities in the same brain regions that incorporate other important aspects of value, specifically category information. This opens the door to future investigations on the neurocomputational mechanisms of integrating different sources of information to guide behavior optimally.

## Methods

**Participants.** In total 29 healthy right-handed volunteers were recruited for the fMRI task. All participants were screened for the use of psychotropic medications, alcohol and drug use, and history of psychiatric disorders, cognitive/neurological disorders, and traumatic brain injury. Data from eleven participants were excluded from further analysis: seven had excessive head motion ( > 2 mm) during the fMRI scan, three did not complete the task due to technical issues, and one participant's verbal report following the scan indicated lack of task understanding. The final sample included 18 healthy right-handed volunteers (12 males) between 18 and 45 years of age (mean $31.1 \pm 8.8$ S.D.). The experiment was approved by Yale School of Medicine Human Investigation Committee. All participants gave informed consent and were paid for their participation.

**Picture rating task and shock tolerance threshold procedure.** Before the main task, participants completed a picture rating task and a shock tolerance threshold procedure. Each participant rated the pleasantness of 20 male and 20 female face images (black-and-white photographs of professional models chosen from the International Affective Picture System[49, 50,]) on a 1–9 scale, with 1 corresponding to "the most unpleasant", and 9 corresponding to "the most pleasant". Following this initial rating, all faces rated as pleasant (6 or above) were presented and the participants were asked to rank them in order of pleasantness. This was done separately for male and female faces. In the case of ties, participants were prompted to rank their preferences among those faces, such that the final ranking was based on strict preferences. Participants were asked if they preferred to see male or female faces during the experiment. Based on the participant's gender preference, the faces at the top, bottom, and the two tertile points of the participant-specific ranking

were chosen. Therefore, in the main task, each participant had a unique set of four faces to be used as pleasant face rewards, based on her/his own preference.

For the shock tolerance threshold procedure, electric shocks of 2 ms duration were administered via electrodes attached to the right ankle. During the threshold procedure, shocks increased in intensity (~ 3 s inter-stimulus-interval) from mild (imperceptible; 10 V) to more intense (up to a maximum 100 V) at 10 V intervals. The participants were asked to report when he or she first felt the shock, and when the shock was "very unpleasant" (but not painful) and he or she would not like it to increase anymore. At this point the threshold procedure was terminated and the upper threshold served as the maximum shock that the participant would receive during the subsequent task. In total 90%, 80%, and 70% of this maximal level would serve as the three lower levels of shock in the main task.

**fMRI paradigm.** In the fMRI paradigm, subjects were presented with a series of four colored shapes, each signaling one of four different categories of rewarding or aversive outcomes—winning money, losing money, viewing a pleasant face, or getting an electric shock (Fig. 1a). These outcomes included both primary (pleasant face images and electric shocks) and secondary (monetary gains and losses) modalities. The meaning of each shape was fully explained to the participants and tested by a few quiz questions and practice trials before the experiment. Each shape was filled up to one of four different levels to indicate the amount or intensity of reward or punishment the participant may receive (Fig. 1b). Upon seeing the cue, the participants were prompted to rate how pleasant they felt in anticipation of the cued outcome. Approximately 1/3 of the trials were followed by the corresponding reward or punishment. Regardless of whether the predicted reward or punishment was delivered or not, the participants were then prompted to rate again how pleasant they felt after seeing the outcome. All cues and outcomes were programmed into the script in advance, and the outcomes did not depend on the responses by the participant.

For actualized money trials the participants could earn or lose $1, $5, $25, or $100. The loss or gain was added to or subtracted from a running tally (which was not displayed to the participant). Each participant was endowed with $20 prior to the start of the task to make sure that they would not lose their own money (the task was pre-programmed such that participants could not lose more than $20). For actualized shock trials, the participants could receive a shock at their ankles with the intensity of 100%, 90%, 80%, or 70% of the participant's discomfort threshold determined by the procedure described above. The duration of shocks of all levels was 2 ms. For actualized pleasant face trials, participants were presented with one picture of his or her own set of four pleasant pictures, depending on the reward level of that particular trial.

The timeline of a representative trial is presented in Fig. 1c. Each trial began with a cue presentation. Participants had to provide their pleasantness rating for the cue within 5.5 s. Ratings were indicated using three buttons on a response box with the index, middle, and ring fingers of the right hand. Two buttons moved a cursor to the left and to the right along the 1–9 scale, and the third button was used to register the selection. A brief delay period of 0.5 s and a presentation of the outcome for 2 s then followed. For actualized shock trials, a 2 ms electric shock of the specified level was delivered during this time. For non-actualized trials of all categories, the message "No Outcome" appeared on the screen. Participants then had another 4 s to rate the outcome pleasantness. The trial length was held constant at 12 s, and the inter-trial interval was jittered between 10 s and 12 s. This slow event-related design was chosen to ensure that responses to each event could be captured. Four blocks of 24 trials each were completed, resulting in a total of 96 trials for each participant. To ensure that participants would not know the total amount of money they would earn before the experiment, one additional monetary gains/losses trial was added to the end of the 4th block, which was randomized among participants, thereby creating different total payoffs for different participants.

After the experiment, a short quiz was administered to ensure that participants understood the task. Depending on the final tally, participants were either paid additional money (up to $45), or had to pay the money they lost during the experiment, out of their $20 endowment. A separate participation fee for the participants' time was also provided.

**Neuroimaging data acquisition and preprocessing.** Participants were scanned in a 3T Siemens Magnetom Tim Trio scanner, using a 12-channel receiver array head coil. High-resolution, T1-weighted anatomical images were collected for each subject using an MPRAGE sequence (TR = 2.5 s, TE = 3.93 ms, TI = 900 ms, flip angle = 8°, 176 sagittal slices, $1 \times 1 \times 1$ mm, $256 \times 256$ matrix in a 256 mm field-of-view, or FOV). Functional data were collected using a standard gradient echo EPI sequence (TR = 2 s, TE = 20 ms, flip angle = 80°, 40 near axial slices at an orientation of 30° to the AC-PC axis, $3 \times 3 \times 4$ mm³, $64 \times 64$ matrix in a $192 \times 192$ mm² FOV) and local shimming to the field of view. Analysis of the imaging data were conducted using BrainVoyager QX Version 2.8, NeuroElf V1.1 software packages (http://www.neuroelf.net), the RSA toolbox[51], and additional in-house Matlab functions. Functional imaging data preprocessing included discarding the first 16 volumes, motion correction, slice scan time correction (using sinc interpolation), spatial smoothing using a three-dimensional Gaussian filter (6 mm FWHM), voxelwise linear detrending, and high-pass filtering of frequencies above three

cycles per scan. Structural and functional data of each participant were then transformed to standard Talairach stereotaxic space[52].

**Behavioral data analysis.** A three-way repeated measures ANOVA was used to determine the effect of outcome valence (reward vs. punishment), modality (primary vs. secondary/monetary), and level on pleasantness ratings across participants. Outliers were detected using boxplots of mean pleasantness ratings for cues predicting outcomes of different levels from different categories (averaging all ratings for trials of the same level of magnitude/intensity from the same category within a participant). There was one outlier in the data, assessed as a value (the mean rating for a cue indicating $5 gain for one participant) greater than 1.5 inter-quartile range from the edge of the box. Mean pleasantness ratings, after removing the outlier, were normally distributed, as assessed by Shapiro–Wilk's test of normality (all $p > .05$). Mauchly's test of sphericity indicated that the assumption of sphericity had been violated for the three-way interaction ($\chi^2_{(5)} = 30.563$, $p = 0.000012$) and the two-way valence × magnitude and modality × magnitude interactions ($\chi^2_{(5)} = 11.122$, $p = 0.049$ and $\chi^2_{(5)} = 12.546$, $p = 0.028$). Therefore, Greenhouse–Geisser correction was applied to the degrees of freedom of $F$ statistics and thereby the assessment of significance of corresponding results.

**fMRI data analysis: general linear models.** Univariate analysis of the neuroimaging data was based on GLMs using ordinary least squares. The main GLM was designed to identify univariate value and saliency signals with all categories pooled together. In this GLM, each trial was divided into two periods[1]: the cue period in the first half (0–6 s) of a trial[2]; the outcome period in the second half of a trial (6–12 s). As we are primarily interested in the anticipation period, the cue period is the focus of this article. To ensure that activation to the delivered outcomes did not contaminate the neural responses to the cue, cues of actualized and non-actualized trials were modeled separately, and only analysis of the cues of non-actualized trials is reported. For non-actualized trials, the cue was modeled by a binary regressor and two parametric regressors modulated by trial-by-trial value and saliency estimates for each participant. Value was defined as the pleasantness rating in the cue period of the current trial; saliency was computed by taking the squared difference of the pleasantness rating and the neutral point 5. For actualized trials, the cue was modeled by a binary regressor only, serving as a regressor of no interest. For future analysis not described in this article, we also included separate binary predictors for actualized and non-actualized outcomes of all categories. Actualized outcomes were further modeled by two parametric regressors modulated by value and saliency of the outcomes. Value and saliency estimates were demeaned prior to creating the parametric regressors. Six motion parameters were included as regressors of no interest, as well as regressors of missed trials (if any). All regressors were convolved with a standard canonical hemodynamic response function. Activation during inter-trial intervals served as baseline.

In a whole-brain single-subject analysis, the model was independently fit to the activity time course of each voxel, yielding eight coefficients for each participant (the presentation of cues of non-actualized trials, cue value of non-actualized trials, cue saliency of non-actualized trials, the presentation of cues of actualized trials, the presentation of outcomes of actualized trials, outcome value of actualized trials, outcome saliency of actualized trials, the presentation of outcomes of non-actualized trials). These coefficients were taken to a random-effects group analysis, in which one-sample $t$ tests over the single-subject contrasts were conducted. A per-voxel threshold of $p < 0.005$ was used, and cluster-size thresholding (at the level of $p < 0.05$) was performed using the cluster-level statistical threshold estimator plugin of the BrainVoyager software. This implementation involves a Monte Carlo simulation of the random process of image generation, taking into account the estimated smoothness of the map, as was described in previous publications[53, 54,]. For the reporting of activation clusters, anatomical locations were determined via visual inspection and the Talairach Daemon database.

A number of other GLMs were constructed for the examination of the encoding of category identities and for control analyses of the saliency signals (see Supplementary Methods for details for all GLMs).

One more GLM was estimated to extract parameter estimates (Fig. 2) from regions of interest that were used to visualize value or saliency representations. In this GLM, each pleasantness rating level from 1 to 9 in the cue period was modeled by a separate binary regressor. It also contained regressors of no interest including cue presentations for actualized trials, outcome periods, missed trials, and motion parameters.

ROI analysis was conducted in two types of ROIs. First, we used external ROIs based on a meta-analysis of human neuroimaging studies on value representation[20]. These ROIs were taken directly from the meta-analysis and are available on the authors' website (http://www.psych.upenn.edu/kable_lab/Joes_Homepage/Resources.html) (Fig. 4). Second, to examine effects of both value and saliency in value/saliency brain areas (Figs. 2 and 3), we defined ROIs using the LOSO procedure[24, 55,] in order to prevent circularity and to avoid the introduction of biases[56]. In the LOSO procedure, we first identified interim ROIs (for either cue value or cue saliency) in certain brain regions from the data of $n$−1 participants. The interim ROIs were defined by carrying out one-sample $t$ tests over the single-subject contrasts statistics using a cluster-size thresholding (1000 samples) with the statistical threshold of FWE $p < 0.05$ (per-voxel threshold $p < 0.005$). New spherical ROIs were constructed around the center of gravity of these interim ROIs with 5

mm radius, and these spherical ROIs were used as the participant-specific ROIs for the independent participant that was not included in the analysis to determine the ROI. We repeated the same procedure n times for each participant using data from the remaining $n$−1 participants. As a result, all such ROIs were determined from an independent sample of participants, thereby avoiding potential biases and so called "double-dipping"[57].

For reporting results from the ROI analyses, mean beta values (regression coefficients/parameter estimates averaged across participants) from the GLMs are plotted (Figs. 2b, c, 3b, c, and 4, Supplementary Figs. 3 and 5). The units of the beta values are on the same scale as z-scores because the BOLD data was z-transformed before fitting the GLM. These β values were then subject to subsequent second-level statistical tests, such as regressions against variables of interest, paired $t$-tests between conditions, or $t$-tests against zero.

**fMRI data analysis: representational similarity analysis.** One participant was excluded for this part of the analysis, because he did not encounter the entire set of 16 conditions in the cue period, due to a technical issue. The following pattern analysis was performed on data from the 17 remaining participants. Data pre-processing for multi-voxel pattern analysis (MVPA) followed a similar procedure to the one used for the univariate analysis, except that no spatial smoothing was performed to preserve the voxelwise activation patterns. To prepare the data for MVPA, a different GLM was fit to the activity time course of each voxel. This GLM included one binary regressor for the cue period of each of the 16 non-actualized trial types (4 categories × 4 levels per category), as well as regressors of no interest for the outcome period and the actualized trials and motion parameters. Voxel-wise activity patterns (β maps) for each of the 16 non-actualized trial types (4 categories × 4 levels per category) were then extracted with this GLM and served as a basis for the analysis described below.

To examine the encoding of value, saliency, and category information in multi-voxel patterns, we used an analysis technique referred to as RSA[58]. This analysis focuses on similarities and dissimilarities in activation patterns for different experimental conditions and examines theoretical models that may explain the observed patterns. This methodology was combined with a whole-brain "searchlight" procedure[59], in which we examined patterns in the immediate neighborhood of every voxel (a 27-voxel cube with that voxel as the center) in the brain. Similar to a whole-brain univariate GLM analysis, the searchlight RSA procedure allowed us to find where in the brain certain task-related variables were encoded, except that now we focused on the multi-voxel pattern of activation, rather than the activation magnitude in single voxels.

The activation patterns of an m-voxel searchlight for all 16 conditions of interest in the task can be represented by 16 m-dimensional vectors of beta coefficients $\mathbf{b_i}$, $i = 1, 2, \ldots, 16$. Using Pearson correlation distance, the dissimilarity between the patterns of two conditions $\mathbf{b_j}$ and $\mathbf{b_k}$ is then quantified as

$$d_{Pearson}(\mathbf{b_j}, \mathbf{b_k}) = 1 - corr(\mathbf{b_j}, \mathbf{b_k}) = 1 - \frac{<\mathbf{b_j}, \mathbf{b_k}>}{\|\mathbf{b_j}\|\|\mathbf{b_k}\|}.$$

One important property of the Pearson correlation distance is that it is invariant to changes in the scaling of elements of $\mathbf{b_i}$, i.e., the magnitude of voxel-wise activities. Therefore, Pearson correlation distance is sensitive to the representational geometry only and can be regarded as a complement to traditional univariate GLM-based analysis of fMRI data, which is based on mean activations. Importantly, Pearson correlation distance does depend on the implicit baseline estimate of the GLM that is used to extract beta values for the various conditions.

For each searchlight, we generated a 16 × 16 neural representational dissimilarity matrix (neural RDM) for each individual participant, based on the Pearson correlation distance between activity patterns for all possible pairs of conditions (Fig. 5).

Candidate model representational dissimilarity matrices (model RDMs) were constructed under different hypotheses of information coding by neural activation patterns. For instance, if category identity of the cues is the only encoded parameter, then the distance between two conditions from the same category will be zero, while the distance between two conditions from different categories will be positive. Similarly, if we hypothesize that a continuous variable (e.g., value or saliency) is represented by multi-voxel patterns of a certain brain region, the distance between any two conditions should be the difference between the means of that variable in these two conditions.

Given the focus of this study on category, value, and saliency encoding, the following candidate models were constructed:

(1) The category model (Fig. 5b). In this model, any two conditions had a distance of 0 if they were from the same category, and 1 if they belonged to two different categories.

(2) The value model (Fig. 5c). The distance between a pair of conditions was determined by difference between the mean values (mean pleasantness ratings) of the two conditions on an individual-subject basis.

(3) The saliency model (Fig. 5d). The distance between a pair of conditions was determined by the difference between the mean saliencies of the two conditions on an individual-subject basis.

Five additional control models for category encoding were examined, which included the following:

(1) The category-mean-value model (Fig. 6a). In this model, the distance between any two conditions from the same category was 0. The distance between any two conditions from two different categories was defined as the difference in mean values (ratings) of the two categories.

(2) The primary-secondary model (Fig. 6b). This model distinguished conditions by their modality only. Therefore, if two conditions were both from the same modality (both primary outcomes, i.e., pleasant faces or electric shocks or both secondary/monetary outcomes), they were indistinguishable by this model, namely their distance was defined as 0. Otherwise, the distance between the two conditions was 1.

(3) The positive-negative model (Fig. 6c). This model distinguished conditions by their valence only. Therefore, if two conditions were both positive outcomes (rewards, i.e., monetary gains or pleasant faces) or both negative outcomes (punishments, i.e., monetary losses or electric shocks), then they were indistinguishable by this model, namely their distance was defined as 0. Otherwise, the distance between the two conditions was 1.

(4) The money vs. face vs. shock model (Fig. 6d). This model was similar to the category model which distinguished conditions by the four categories, except that here monetary gains and losses were combined as one single category.

(5) The monetary gains vs. losses vs. primary model (Fig. 6e). This model was similar to the category model which distinguished conditions by the four categories, except that here the two primary categories (pleasant faces and electric shocks) were combined as one single category.

Neural RDMs (to be explained) were then compared to candidate model RDMs (serving to explain). Statistical inference was performed to assess the contribution of each candidate model RDM to the neural RDM, using the RSA toolbox[51] and customized Matlab scripts. The first-level analysis was conducted at the level of individual participants; Spearman's rank correlation coefficients between the single-subject neural RDM and the candidate model RDM of interest for this participant were computed, using only the lower triangles (excluding the diagonal elements) of both RDMs. The Spearman's rank correlation coefficients of all participants were then taken to a second-level group analysis, in which a one-sided signed-rank test across the single-subject correlations was performed, generating the effect size estimate ($z$-value) for this candidate model. For searchlights, each $z$-value was assigned to the searchlight's center voxel. Whole-brain $z$-maps formed by this procedure were then subject to statistical thresholding on a per-voxel basis (at the level of $p < 0.005$) and cluster-size thresholding (at the level of FWE $p < 0.05$ with 1000 random samples) using the NeuroElf package.

**Principal component analysis**. To better elucidate the nature of neural signals in the form of multi-voxel patterns, PCA were run on the β coefficients for the 16 cue conditions in various ROIs. The suitability of PCA (including multivariate normality, linearity, and moderate correlations between dimensions) was assessed prior to analysis. For each brain region, the PCA was first conducted on the β maps of this brain region on every participant. The percentages of total variance explained by the first 5 principal components in each participant are presented in Fig. 7a. The first 3 principal components were used in mixed-effects regressions, in which our main variables of interest (category identity, value, and saliency) served as independent variables and subject was treated as a random effect. Category identity was coded by three dummy variables for monetary gains, monetary losses, and pleasant face, respectively, and the electric shock served as the reference category. Group level statistics were reported for the significance of the effects of category, value, and saliency on each of the principal components. To assess the loading structures of PCs and the consistency across subjects, we first calculated the frequency of loading coefficients falling under a particular bin separately for each participant, and obtained both the mean and the S.E.M. of frequencies in this bin across participants, and then repeated this for all bins, spanning the entire range of loading coefficients. We then plotted the mean frequencies and the S.E.Ms in each bin in the form of a histogram with error bars, as shown in Fig. 7b and Supplementary Fig. 7b.

**Code availability**. Analysis codes for RSA and PCA are available at https://github.com/zhihao13/Zhang_et_al_17.

**Data availability**. All relevant data and other codes are available from the authors upon reasonable request.

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

## Acknowledgements

This work was funded by NIH/NIMH grant R21MH102634 and CTSA grant (UL1 TR000142) from the National Center for Advancing Translational Science (NCATS) to I.L. We thank Daniela Schiller, Timothy Vickery, and Xiaodi Hou for very helpful discussions, and Eric Jackson for technical help.

## Author contributions

J.F., D.L., and I.L. designed study, J.F. and Z.Z. conducted study, Z.Z., W.C., and D.B.E. analyzed data, Z.Z. and I.L. wrote manuscript, all authors commented and approved the final manuscript.

## Additional information

**Competing interests:** The authors declare no competing financial interests.

