## [Peer Review File · Nature Communications]

Reviewers' comments:

Reviewer #1 (Remarks to the Author):

Zhang and colleagues report a study of the neural representations of value using functional magnetic resonance imaging (fMRI) in a challenging and rare combination of both primary and secondary rewards and physical punishments. Using a task paradigm that supports direct comparison across the outcomes utilized, the authors use a series of analyses to disentangle the relative contributions of value, salience, and "category". The authors describe distributed representations of both value and salience with some integration across category type in areas previously associated with common currency descriptions.

The paper fills an empty niche in an area that has been heavily studied. I find the methods and results compelling and potentially of relatively broad interest. Please address the following larger concerns:

1. The use of category is confusing (category descriptions in the introduction is a nice microcosm). For the examples provided, circumstance or context is a better descriptor and has a large relevant literature to aid interpretation. The authors should work to improve the terms and descriptions of "category" to ensure it matches the operationalization and .
2. It seems like there is a potential mean bias in using a predefined neutral set point when salience is defined using ratings. This would change the interpretation of the salience results.
3. The results are described largely as a mapping experiment without ties to base models of the neural computation of value.
4. The analysis is complex enough that it is difficult to review in the manuscript format. To aid replicability and assist further review as the manuscript is read, please post your scripts in a repository like GitHub.

Minor

- + Expectancy and pleasantness seem to be used interchangeably. Consistent use of a term would help. If they are being used to indicate two different measures, when and why expectancy is used instead of pleasantness should be better clarified throughout. In particular which metric is being used to support a particular claim. Axis labels could also be improved.
- + Pg 5 line 111 - "...clarify existing ambiguities..." - sentence is unclear and may need to be unpacked.
- + Pg 7 line 146 sentence starting "The effect of value.." could be more simply/clearly written.
- + Page 8 line 173 - marginal test described as not significant.
- + Page 9 line 192 - small area is not located by...
- + Check results for interpretations and move those to the discussion.
- + Page 14 line 303 - "this implies..." requires model comparison (report statistic). Not significant in one model does not mean the two are significantly different.
- + Page 16 line 355 - T's reported for specific item versus all others? Please clarify.

Reviewer #2 (Remarks to the Author):

In this paper the authors use fMRI and a paradigm crossing “outcome valence” (reward/punishment) with “outcome type” (monetary gains and losses, and pleasant faces/shocks) to attempt to separate the neural underpinnings of value, saliency, and category. This remains an important category of question for work in human neuroscience especially during a time when these variables are 'reported' by many papers, but are often not clearly defined and rigorously tracked with respect to some neural probe of choice.

The authors employ a range of approaches including univariate, multivariate, RSA (representational similarity analysis) and PCA (principal components analysis).

Summary -

The univariate results for value and salience (using models with parametric regressors for both value and salience) are straightforward and perhaps not surprising. The results for the univariate category analysis were mostly confined to visual areas.

The results in the non-visual areas trended towards a category specific representation, but a conjunction analysis was largely not positive.

The RSA analysis involved comparing subjects' representational dissimilarity matrix (RDM; constructed at a particular voxel from the correlation distance of the beta maps for the 16 categories (4 levels for each of the 2x2 categories) over 27 nearest voxels) with various RDMs models based on various hypotheses about how things ought to look if the representation is about category, value, etc. Each of these presented assumptions ultimately yields a whole-brain map that shows how well (as a group) the RDM matches the model (for a voxel). The map for the category model yields a widely distributed map (including overlaps with value regions, vmPFC and striatum) of significant values, while the maps for the other models tend to be more localized. Finally, the authors use a PCA analysis of the beta coefficients of the 16 “cue conditions” in an ROI in vmPFC to examine the interaction between the univariate and multivariate representations. The main result is that the loadings of the first PC (for vmPFC) correlated significantly with value (but not saliency or category), the second had no significant correlations, while the third correlated only with category.

Overall Impression

This is a thorough exploration of issues surrounding valence x modality, and the authors have done a voluminous amount of careful work. The univariate evidence for valence and salience is quite solid, and even though some of it may overlap with previous results, it is a significant contribution. The supplemental analysis of “value salience” and “visual salience” is appreciated, as is the button-press control. The univariate category results are less impressive. This claim is supported by the fact that there are no main figures on them. The multivariate results are more exploratory in nature.

The dissimilarity matrix results were interesting and I think that they will be to the wider readership. I felt that the PCA results were not adequately explained for the general reader.

Major Points

The PCA result is not well motivated; nor are the methods of this part well-explained for the general reader.

Please spell out exactly what was done. For example it is unclear how different % variance explained numbers were obtained by subject. In the text it says: "For each brain region, the PCA was first conducted on the beta maps of this brain region on every subject." Does the mean over all conditions over all subjects? Or does it mean you did it by subject? If it is by subject, do the PCs across subjects look the same? If you did it with all subjects' data pooled, please state how the % variance explained numbers (by subject) were obtained.

Minor Points

1. The models used for in the univariate analysis of valence and salience for the whole brain are well-described in the Supplemental Methods, but the paper would be easier to read if the basics of the model are referred to in the main methods, and even in the legends (the fact that both parametric modulators are in the model).

2. I could have missed the explanatory bits here but the ROI analysis was difficult to follow. Was a two-step procedure used as in the whole-brain analysis (estimate the model by subject separately then t-tests on the mean)?

OR

Was a model including all the subjects estimated?

This left me wondering exactly how the ROIs were handled.

3. It would be interesting to see (in the supplement or perhaps a separate small paper) separate analyses for the two modalities (across loss and gain).

Reviewer #3 (Remarks to the Author):

The authors present an fMRI study in healthy subjects examining neural representation of the value of anticipated positive and negative stimuli. Building upon previous work, they aim to avoid a potential confound in previous studies on this topic, in which stimulus salience could be conflated with stimulus value if the stimuli all have the same valence - by examining stimuli of positive and negative valence, they aim to dissociate representation of salience from representation of value.

They identify lateral orbitofrontal and posterior parietal regions more sensitive to value, distinct from a network of rostral ACC, anterior insula, and ventral striatal regions more sensitive to salience. In addition, they apply a searchlight whole-brain multi-voxel pattern analysis approach aimed at identifying regions sensitive to stimulus value, salience, and category.

The more rigorous examination of value coding, with control of the potential confounding effects of salience, is attractive. The authors' efforts to rule out several additional potential confounds in the supplementary material are also an attractive feature. On the whole, the results of the GLM analyses identifying distinctive value- and salience-coding regions are more compelling than the results of the RSA analyses, which appear to turn up a less well-demarcated array of regions dominated in many cases by modality-specific areas in occipital cortex.

The paper is well written and from a technical standpoint appears well executed. The number of

subjects was relatively low, although it does appear that the authors were able to extract enough signal from this sample to distinguish value and saliency representations. This is an important, if not a transformative, advance over previous work in the field.

The RSA analyses aimed at identifying category representation appear less successful, and the paper's impact might be improved by focusing more on the analyses saliency / value dissociation analyses. The array of stimuli presented in this experiment appears better suited to a rich mapping of saliency vs value, with a less rich mapping of stimulus categories. The authors make a good point in their Introduction that the calculation of value requires both information about the organism's internal state and information about the properties of the stimulus under consideration. One might develop this point further to consider value as a sort of dot product of a vector describing the organism's current homeostatic state [caloric balance, osmolar balance, thermal balance, etc etc] with a corresponding vector describing the homeostatic properties of the stimulus under consideration.

If the latter can be thought of as a 'category representation' for the purposes of value calculation, then previous literature (e.g. Ref 1 below) does point to anterior temporal lobe regions as representing these homeostatic properties of stimuli as a source of input to the anatomically connected orbitofrontal regions that the authors have nicely demonstrated are calculating value.

From this perspective, a rather different experiment, with a richer array of stimulus types and taking place under different conditions of homeostatic deprivation, would be needed to properly explore the contributions of 'category' (i.e., homeostatic properties of a stimulus) to value calculation. As such, the current paper's impact might be improved by focusing on the rather well-executed dissociation of saliency from value, rather than the less compelling examination of the how stimulus properties are represented during value calculations.

With this major revision, the paper may be worth re-consideration for publication in this journal.

Some other minor suggestions for revision.

- In Table 1 - an activation is labelled as "middle frontal gyrus/lateral orbitofrontal cortex" - these regions are distant from one another and the supplied Talairach coordinate seems to fall within OFC, quite ventral to MFG. Could the authors clarify the reason for labelling this region as MFG?

- In Figs 1-3: Y-axes declare that Z-coefficients are presented, but if they are Z-scores, then values are quite low (0.1-0.4 in many cases) for the significance levels given. Is it possible that these are actually raw R-values prior to transformation to Z-values? Could the authors please clarify?

1. Rice GE, Lambon Ralph MA, Hoffman P (2015): The Roles of Left Versus Right Anterior Temporal Lobes in Conceptual Knowledge: An ALE Meta-analysis of 97 Functional Neuroimaging Studies. *Cerebral cortex*. 25: 4374-4391

We thank the reviewers for the positive evaluation of our manuscript and for the very helpful constructive comments. Below we describe the revisions we have made in response to the detailed comments.

Reviewer #1 (Remarks to the Author):

Zhang and colleagues report a study of the neural representations of value using functional magnetic resonance imaging (fMRI) in a challenging and rare combination of both primary and secondary rewards and physical punishments. Using a task paradigm that supports direct comparison across the outcomes utilized, the authors use a series of analyses to disentangle the relative contributions of value, salience, and “category”. The authors describe distributed representations of both value and salience with some integration across category type in areas previously associated with common currency descriptions.

The paper fills an empty niche in an area that has been heavily studied. I find the methods and results compelling and potentially of relatively broad interest. Please address the following larger concerns:

1. The use of category is confusing (category descriptions in the introduction is a nice microcosm). For the examples provided, circumstance or context is a better descriptor and has a large relevant literature to aid interpretation. The authors should work to improve the terms and descriptions of “category” to ensure it matches the operationalization and .

We apologize for the confusion caused by our previous examples, and we thank the reviewer for the opportunity to clarify it here and in the manuscript. Throughout the study, we used ‘category’ to refer to the classification of rewarding/punishing outcomes based on their nature, e.g. whether a reward belongs to monetary gains, food, water, pleasant scenes, physical comfort, social reward, etc. Therefore, category is a stable attribute of an outcome regardless of circumstance or context, and this term has been used in a large number of previous studies, although almost all of them did not pertain to valuation and decision-making (except McNamee et al., 2013).

In the Introduction of our previous submission, when we used the example of the valuation of an ice cream “after workout on a hot sunny afternoon” vs. “when dieting to lose weight on a cold winter day”, we meant to illustrate that category is an important factor to determine whether value needs to be updated when circumstances change. Clearly, an ice cream is usually more valuable in the former than the latter circumstance, while some other outcome (e.g. money) should have similar value in both cases. However, circumstance/context is outside of the scope for this study, and it can

be seen from the rest of the manuscript that we did not manipulate circumstance/context in our experiment.

We completely agree with the reviewer that, without sufficient explanation, it was unclear to readers what we were referring to as 'category'. We have revised this part of the Introduction as follows (page 4):

“The representation of category identity has primarily been the subject of studies on semantic memory and conceptual knowledge, with little consideration of valuation (17-19). However, information about the nature and category of each option is integral to making optimal decisions. Importantly, it is required not just for the initial computation of value, but also for subsequent updates. For example, the same ice cream may be more valuable on a hot compared to a cold day, while the value of a \$10 bill should not change much in different weather. Maintaining category information allows integration of changes in the external environment and one’s internal state into the update process. We hypothesized that category information may coexist with value signals to facilitate their flexible integration.”

2. It seems like there is a potential mean bias in using a predefined neutral set point when salience is defined using ratings. This would change the interpretation of the salience results.

The reviewer is wondering if using a predefined neutral set point (the rating 5) could lead to a potential mean bias. An alternative approach would be some kind of normalization (i.e. z-scoring), which could in principle take into account individual differences in how subjects use the rating scale. Normalization of data on a per-subject basis consists of two steps: 1) centering the data by subtracting the mean rating of the subject; 2) rescaling the centered data by units of standard deviation of the ratings of the subject. Below we discuss the implications of each step and how our approach compares to normalization.

For the calculation of saliency, centering the data causes the mean rating for each subject to become the neutral point for them. This is a strong assumption that we feel unwarranted. First, as explained in the manuscript, when we explained the rating scale to the subject during the pre-scan instructions, we did provide a standard interpretation of the ratings by emphasizing that 5 was neutral, and anything above/below 5 would be pleasant/unpleasant (these interpretations also appeared on the screen next to the scale in every trial). Admittedly, there is no objective way of verifying that all subjects used the scale in the way we asked them to, but it seems appropriately conservative to assume so without further information of a systematic bias. Second, using the mean rating as the neutral point also implicitly assumes that overall the positive and negative outcomes in our paradigm cancelled each other out in terms of deviations from neutrality, which may not necessarily be true. To illustrate this, let us imagine a subject using the rating scale with 5 being her subjective neutral point, who gives monetary

gains, monetary losses, and pleasant faces average ratings of 7, 3, and 7. Meanwhile, this subject is particularly sensitive to pain, so the electric shocks are more negative (and more salient) to her. As a result, her average rating for electric shocks is 1.5 (still with 5 being neutral). Under such circumstance, the mean rating across all categories for this subject will be below 5, but using this mean as the neutral point to calculate saliency would introduce much bias – the saliency for the two negative categories would be lower, while the two positive categories would have inflated saliency.

With such caveat in mind, we can also examine the distribution of subject-level mean ratings in our behavioral data, to search for subjects (if any) who only reported very high or very low ratings (Figure R1A). It can be seen that the subject-level mean ratings were indeed distributed within a relatively narrow range around 5, and the ‘mean of the mean’ was 4.93. This indicates that our dataset did not have extreme cases where only one end of the scale was used (e.g. a ‘pessimistic’ subject using 5 as the highest response).

Figure R1 Subject-level distribution characteristics of cue ratings

A. Histogram of subject-level mean cue ratings across all categories. B. Histogram of subject-level standard deviation of ratings across all categories.

The second step of normalization, rescaling centered data by the standard deviation, essentially means adjusting the saliency by the range of ratings of individual subjects in this context. The distribution of subject-level standard deviations of ratings is presented in Figure R1B, where a moderate degree of individual differences can be observed. It is important to recognize the two potential sources of such differences: individual

differences in the usage of the scale and/or individual differences in stimuli/outcome evaluation. For example, two subjects, A and B, had ratings ranging from 1 to 9 and 3 to 7, respectively. While it is possible that the two of them had the same experience in the task and only differed in how they used the rating scale, it is also possible that subject A did have a more intense affective experience (thereby having a greater dynamic range for ratings) than subject B. In the latter case, assuming that a rating of 9 for A indicates the same level of pleasantness and salience as a rating of 7 for B would be detrimental for examining the neural representation of these quantities across subjects.

To summarize, calculating saliency by normalization (z-scoring) entails making a set of strong assumptions of how ratings map onto subjective value/saliency, which are hard to verify. More broadly speaking, these issues are general limitations of working with ratings. We agree with the reviewer that this is a limitation of using explicit ratings, and do not attempt to argue that our approach is the 'right' one, but it seems to be a more conservative choice given how the experiment was designed and administered, as well as our subjects' behavior. It is important that readers evaluate our results with these caveats in mind. We point this out in the Supplementary Discussion section (Supplementary Materials page 11).

3. The results are described largely as a mapping experiment without ties to base models of the neural computation of value.

We agree with the reviewer that our results should be better described and discussed in the context of models of the neural computation of value. In particular, our findings that category information also exists in brain regions traditionally associated with value representation like vmPFC and VS is consistent with recent studies showing the importance of maintaining and retrieving information about specific attributes about the option being considered. We now include an additional paragraph in the Discussion section to elaborate such links (page 17):

"The results are also of relevance for models of value computation. Prior research shows that vmPFC integrates multiple attribute-value signals from other, specialized, brain areas(39, 40). There is also evidence suggesting that in addition to comparisons of these integrated values, the choice process also entails direct comparisons at the level of single attributes (41). Moreover, recent research suggests that value may be determined, to a large extent, based on memories of individual instances of encountering the particular reward (42, 43). Current models of value and choice thus require the simultaneous encoding of category-independent value and of category information (as well as other specific information of the evaluated option). Future research will need to determine how conceptual category knowledge in temporal regions (18) is transformed into the category information encoded in value areas. Such research will also need to reveal the nature of the multidimensional value signals in vmPFC (and perhaps also the VS), including their regional specificity and context dependence, temporal dynamics of the category information

in value computation, and whether the effects we observed here extend beyond the four outcome categories included in our design.”

4. The analysis is complex enough that it is difficult to review in the manuscript format. To aid replicability and assist further review as the manuscript is read, please post your scripts in a repository like GitHub.

We have posted our scripts for the leave-one-subject-out analysis, the RSA analysis, and the PCA analysis in the following GitHub repository (also listed in the manuscript): https://github.com/zhihao13/Zhang_et_al_17.

Minor

+ Expectancy and pleasantness seem to be used interchangeably. Consistent use of a term would help. If they are being used to indicate two different measures, when and why expectancy is used instead of pleasantness should be better clarified throughout. In particular which metric is being used to support a particular claim. Axis labels could also be improved.

We have revised the descriptions such that ‘pleasantness ratings’ are now used consistently. We have also improved axis labels in Figure 1D and Supplementary Figure 1.

+ Pg 5 line 111 - “...clarify existing ambiguities...” - sentence is unclear and may need to be unpacked.

We thank the reviewer for pointing this (and the next one) out. We have revised this sentence, and it now reads as follows:

“These findings help clarify existing ambiguities regarding ‘common-currency’ value encoding in the human brain, and provide new insights on how category information may be integrated with value signals.”

+ Pg 7 line 146 sentence starting “The effect of value..” could be more simply/clearly written.

We have revised this sentence, which now reads as follows:

“To visualize the effect of value on activation in these regions, we extracted parameter estimates, determined by the leave-one-subject-out (LOSO) procedure, and plotted them as a function of pleasantness ratings for the cues (Figure 2 middle).”

+ Page 8 line 173 - marginal test described as not significant.

We have revised our description of this result and made it more clear that one of the tests was marginally significant:

“The activities of these ROIs also demonstrated statistically significant saliency effects (Figure 3 bottom; $t_{(17)} = 5.01$, $p = 0.00013$ for rACC; $t_{(17)} = 3.66$, $p = 0.0021$ for striatum; $t_{(17)} = 5.14$, $p = 0.0001$ for AI), but not value ($t_{(17)} = 1.26$, $p = 0.23$ for striatum; $t_{(17)} = 1.18$, $p = 0.26$ for AI), except for a marginally significant effect in the rACC ($t_{(17)} = 1.84$, $p = 0.084$).”

+ Page 9 line 192 - small area is not located by...

We thank the reviewer for catching this and have revised this sentence:

In addition, a small area in the VS was also identified in a whole-brain conjunction analysis of correlation with cue value and with cue saliency (center Talairach coordinates $X = -6$, $Y = 2$, $Z = -2$), if a less stringent statistical threshold were used (per-voxel $p < 0.005$ uncorrected, cluster size > 20 voxels).

+ Check results for interpretations and move those to the discussion.

In this revision, we have moved most interpretations in the Results section to the Discussion section. Occasionally, we feel that a brief, one-sentence interpretation is necessary to motivate the analysis that follows and create a coherent narrative. Therefore, we are keeping a minimum of interpretations in the discussion. We will gladly remove these sentences, however, if the reviewer thinks we should.

+ Page 14 line 303 - “this implies...” requires model comparison (report statistic). Not significant in one model does not mean the two are significantly different.

We have removed this sentence and rephrased based on the reviewer’s comment.

+ Page 16 line 355 - T’s reported for specific item versus all others? Please clarify.

We apologize for not being clear in describing these statistics. In these principal component regressions, category identity is coded by three dummy variables, with the electric shock as the reference category. Therefore, effects of category identity are represented as specific categories compared to the electric shock category. We have clarified this in the revised manuscript:

“The first PC was significantly correlated with value ($t_{(16)} = 2.44, p = 0.027$), but not with saliency ($t_{(16)} = 0.98, p = 0.34$) or category (coded as three dummy variables, using the electric shock category as the reference category...

... while PC3 showed trends towards significance for value ($t_{(16)} = 1.805, p = 0.09$) and category (compared to electric shocks: monetary gains $t_{(16)} = 2.304, p = 0.035$; monetary losses $t_{(16)} = -0.064, p = 0.95$; pleasant faces $t_{(16)} = 1.801, p = 0.09$).”

Reviewer #2 (Remarks to the Author):

In this paper the authors use fMRI and a paradigm crossing “outcome valence” (reward/punishment) with “outcome type” (monetary gains and loses, and pleasant faces/shocks) to attempt to separate the neural underpinnings of value, saliency, and category. This remains an important category of question for work in human neuroscience especially during a time when these variables are 'reported' by many papers, but are often not clearly defined and rigorously tracked with respect to some neural probe of choice.

The authors employ a range of approaches including univariate, multivariate, RSA (representational similarity analysis) and PCA (principal components analysis).

Summary -

The univariate results for value and salience (using models with parametric regressors for both value and salience) are straightforward and perhaps not surprising. The results for the univariate category analysis were mostly confined to visual areas.

The results in the non-visual areas trended towards a category specific representation, but a conjunction analysis was largely not positive.

The RSA analysis involved comparing subjects' representational dissimilarity matrix (RDM; constructed at a particular voxel from the correlation distance of the beta maps

for the 16 categories (4 levels for each of the 2x2 categories) over 27 nearest voxels) with various RDMs models based on various hypotheses about how things ought to look if the representation is about category, value, etc. Each of these presented assumptions ultimately yields a whole-brain map that shows how well (as a group) the RDM matches the model (for a voxel). The map for the category model yields a widely distributed map (including overlaps with value regions, vmPFC and striatum) of significant values, while the maps for the other models tend to be more localized. Finally, the authors use a PCA analysis of the beta coefficients of the 16 “cue conditions” in an ROI in vmPFC to examine the interaction between the univariate and multivariate representations. The main result is that the loadings of the first PC (for vmPFC) correlated significantly with value (but not saliency or category), the second had no significant correlations, while the third correlated only with category.

Overall Impression

This is a thorough exploration of issues surrounding valence x modality, and the authors have done a voluminous amount of careful work. The univariate evidence for valence and saliency is quite solid, and even though some of it may overlap with previous results, it is a significant contribution. The supplemental analysis of “value saliency” and “visual saliency” is appreciated, as is the button-press control. The univariate category results are less impressive. This claim is supported by the fact that there are no main figures on them. The multivariate results are more exploratory in nature.

The dissimilarity matrix results were interesting and I think that they will be to the wider readership. I felt that the PCA results were not adequately explained for the general reader.

Major Points

The PCA result is not well motivated; nor are the methods of this part well-explained for the general reader.

Please spell out exactly what was done. For example it is unclear how different % variance explained numbers were obtained by subject. In the text it says: “For each brain region, the PCA was first conducted on the beta maps of this brain region on every subject.” Does the mean over all conditions over all subjects? Or does it mean you did it by subject? If it is by subject, do the PCs across subjects look the same? If you did it with all subjects’ data pooled, please state how the % variance explained numbers (by subject) were obtained.

We thank the reviewer for this important comment – we were indeed unclear in motivating the PCA and in explaining the detailed procedure. We conducted the PCA primarily for the purpose of using a data-driven approach to examine what information

was represented in the neural activities in important ROIs including the vmPFC and VS. Since PCA is capable of revealing putative signals from the covariance structure of multi-dimensional data without a priori hypotheses, it offers a different perspective to the data complementing the univariate GLM and the RSA analyses. This was discussed in the Discussion section (page 18), but not adequately explained in the Results section of our previous submission. In this revision, we elaborate on the motivations for performing the PCA before describing the results (page 11):

“RSA revealed considerable spatial overlap between regions exhibiting multi-voxel encoding of category information and the well-established valuation areas, including the vmPFC and VS. This suggests an intriguing possibility: category information and value may coexist within the same brain region, but in different forms. To probe this possibility we performed a data-driven analysis using principal component analysis (PCA), a methodology that reveals the internal structure in multidimensional data and does not assume particular forms of the signal. We expected that PCA would uncover both value and category information from the valuation areas, and that the loading structures of the identified neural signals may reveal distinct forms of encoding, consistent with the univariate and RSA findings.”

We also agree with the reviewer that the methods were not well explained. To answer the reviewer’s questions, the PCA was performed over all 16 conditions within each subject, from which we obtained the percent variance explained by subject shown in Figure 7A and Supplementary Figure 6A. However, in our previous submission, the histograms of loading coefficients were generated by taking the average of the loading coefficients for each voxel in the ROI across all subjects, and then plotting the distribution of these average loading coefficients across voxels, which we now realize is problematic. This is because such visualization may fail to preserve information about the overall loading structure of the PCs, which determines the nature (univariate vs. multivariate) of the signals in them. In particular, potential individual differences in the loading structure would be obscured and the aggregated histogram could be misleading – for example, if half of the subjects had high loadings on 50% of the voxels and low loadings on the remaining, while the other half of the subjects had the reverse, plotting the distribution of the average per-voxel loadings might produce a relatively flat distribution inconsistent with any of the subjects.

This is not the case for our study. Examining the individual-subject histograms of loading coefficients confirmed that there is great consistency in the patterns of loading structures as we described. Given the consideration above, we have revised Figure 7B and Supplementary Figure 6B (shown below as Figure R2) so that they present the average distribution (not the distribution of average loadings) together with the variability across subjects. More specifically, we first calculated the frequency of loading coefficients falling under a particular bin separately for all subjects, obtained both the mean and the S.E.M. of frequencies in this bin, then repeated this for all bins, spanning the entire range of loading coefficients. We then plotted the mean frequencies and the S.E.M.s in each bin in the form of a histogram with error bars. It can be seen from the size of the error bars that the individual differences in the distributions of loading

coefficients were small. With such, we hope it is now clear to readers that the loading structures showed a consistent pattern across subjects: PC1 resembled the mean activity of all voxels in the ROI, while PC2 and PC3 reflected more complicated multi-voxel patterns. Note that this does not necessarily mean that the same voxel would have similar loading coefficients across subjects - it is the overall distribution of loading that is consistent in different subjects.

Figure R2. Average distributions of loading coefficients of vmPFC (left) and VS (right).

We have added the following description to the Methods section (page 28):

“To assess the loading structures of PCs and the consistency across subjects, we first calculated the frequency of loading coefficients falling under a particular bin separately for each participant, and obtained both the mean and the S.E.M. of frequencies in this bin across participants, and then repeated this for all bins, spanning the entire range of loading coefficients. We then plotted the mean frequencies and the S.E.M.s in each bin in the form of a histogram with error bars as shown in Figure 7B.”

We have also revised the figure captions accordingly.

Minor Points

1. The models used for in the univariate analysis of valence and salience for the whole brain are well-described in the Supplemental Methods, but the paper would be easier to read if the basics of the model are referred to in the main methods, and even in the legends (the fact that both parametric modulators are in the model).

We agree with the reviewer and have moved most of the methods description to the main Methods section to make it more accessible to readers. We now also emphasize in the captions for Figures 2 and 3 the fact that both the parametric modulator for value and for saliency are included in the same model.

2. I could have missed the explanatory bits here but the ROI analysis was difficult to follow. Was a two-step procedure used as in the whole-brain analysis (estimate the model by subject separately then t-tests on the mean)?

OR

Was a model including all the subjects estimated?

This left me wondering exactly how the ROIs were handled.

We apologize for the confusion. We assume that the reviewer was referring to the ROIs defined using our own dataset, as the use of external ROIs from prior studies was straightforward. To avoid circularity, we used a leave-one-subject-out procedure to define ROIs in the following manner.

The general idea is that, in a dataset with n subjects, the ROI used for the subsequent analysis for any given subject was defined using the data from the remaining $n-1$ subjects, excluding the one in consideration. For example, to define the value ROIs for subject 1, we performed a random-effects analysis (the two-step procedure as mentioned by the reviewer) with the value vs. baseline contrast on the data of subjects 2, 3, 4, ..., 18. The activations evolving from this analysis were then subject to statistical thresholding, and the locations of the centers of gravity were recorded. We then constructed new spherical ROIs around the center of gravity of these activations with 5mm radius, and these ROIs were saved as value ROIs for subject 1. For subjects 2, the same procedure was followed, except that the activations were generated from a random-effects analysis with the value contrast on the data of subjects 1, 3, 4, 5 ..., 18,

excluding subject 2 herself. We repeated this for every subject, obtaining a set of subject-specific value ROIs using samples independent of the subject of interest.

For further analysis, we then extracted parameter estimates for each subject from these subject-specific ROIs using GLMs modeling either pleasantness ratings (Figures 2B & 3B) or value vs. saliency (Figure 2C & 3C).

3. It would be interesting to see (in the supplement or perhaps a separate small paper) separate analyses for the two modalities (across loss and gain).

We wholeheartedly agree with the reviewer that this is a great idea. Unfortunately, we do not have sufficient statistical power for separate analyses in gains and losses in the current design, given that such analyses would reduce the number of trials in half. For the same reason, we do not analyze the neural data for the outcome period or compare them to the cue period in this manuscript, as trials with delivered outcome only constituted one third of the trials. We are currently planning to pursue these questions in a larger study with a modified paradigm.

Reviewer #3 (Remarks to the Author):

The authors present an fMRI study in healthy subjects examining neural representation of the value of anticipated positive and negative stimuli. Building upon previous work, they aim to avoid a potential confound in previous studies on this topic, in which stimulus salience could be conflated with stimulus value if the stimuli all have the same valence - by examining stimuli of positive and negative valence, they aim to dissociate representation of salience from representation of value.

They identify lateral orbitofrontal and posterior parietal regions more sensitive to value, distinct from a network of rostral ACC, anterior insula, and ventral striatal regions more sensitive to salience. In addition, they apply a searchlight whole-brain multi-voxel pattern analysis approach aimed at identifying regions sensitive to stimulus value, salience, and category.

The more rigorous examination of value coding, with control of the potential confounding effects of salience, is attractive. The authors' efforts to rule out several additional potential confounds in the supplementary material are also an attractive feature. On the whole, the results of the GLM analyses identifying distinctive value- and salience-coding regions are more compelling than the results of the RSA analyses, which appear to turn up a less well-demarcated array of regions dominated in many cases by modality-specific areas in occipital cortex.

The paper is well written and from a technical standpoint appears well executed. The number of subjects was relatively low, although it does appear that the authors were able to extract enough signal from this sample to distinguish value and saliency representations. This is an important, if not a transformative, advance over previous work in the field.

The RSA analyses aimed at identifying category representation appear less successful, and the paper's impact might be improved by focusing more on the analyses saliency / value dissociation analyses. The array of stimuli presented in this experiment appears better suited to a rich mapping of saliency vs value, with a less rich mapping of stimulus categories. The authors make a good point in their Introduction that the calculation of value requires both information about the organism's internal state and information about the properties of the stimulus under consideration. One might develop this point further to consider value as a sort of dot product of a vector describing the organism's current homeostatic state [caloric balance, osmolar balance, thermal balance, etc etc] with a corresponding vector describing the homeostatic properties of the stimulus under consideration.

If the latter can be thought of as a 'category representation' for the purposes of value calculation, then previous literature (e.g. Ref 1 below) does point to anterior temporal lobe regions as representing these homeostatic properties of stimuli as a source of input to the anatomically connected orbitofrontal regions that the authors have nicely demonstrated are calculating value.

From this perspective, a rather different experiment, with a richer array of stimulus types and taking place under different conditions of homeostatic deprivation, would be needed to properly explore the contributions of 'category' (i.e., homeostatic properties of a stimulus) to value calculation. As such, the current paper's impact might be improved by focusing on the rather well-executed dissociation of saliency from value, rather than the less compelling examination of the how stimulus properties are represented during value calculations.

With this major revision, the paper may be worth re-consideration for publication in this journal.

We appreciate the reviewer's positive assessment of our manuscript. We agree with the reviewer that a richer array of categories would be very helpful for investigating the representation of category identity. Following the reviewer's suggestions, we have made the following revisions:

(1) We have substantially revised the Introduction section, so that it is more clear to readers that the primary goal of this study was to examine category-general value representations in the brain and to fully dissociate it from the encoding of saliency. Meanwhile, the analysis on category encoding was made possible by our experimental paradigm that included different outcome categories. Because of the relative lack of existing decision neuroscience literature on this topic, this part of the results was more exploratory in nature.

(2) We have removed the results for the univariate category representation from the main text, and only summarized them in the Supplementary Materials for the sake of completeness.

(3) We have also revised the Discussion section, so that it focuses more on the dissociation of category-general value vs. saliency encoding, and less on the representation of category.

(4) We also discuss the relevance of conceptual knowledge to the question of category signals in valuation processes, and have incorporated the reference that this reviewer pointed us to.

We would still like to keep most of the RSA and PCA results in the manuscript for the following considerations:

First, in addition to category, the RSA and PCA also examined the multi-voxel encoding of value, saliency, and some other constructs (Figures 5 & 6). Being one of the first studies using whole-brain RSA in decision neuroscience, we believe that this is an important development from existing studies using other multi-voxel pattern analysis methods (e.g. Kahnt et al., 2014, PNAS) or RSA on specific ROIs (e.g. Jin et al., 2015, J Neurosci). The use of PCA as a data-driven method to separate different types of signals in multi-dimensional data is also a novel contribution, as its application on human fMRI data is rare so far and it offers the unique capability of uncovering univariate and multivariate signals in a unified manner.

Second, although the relatively small number of categories raised the question about the generalizability of our findings, the nature of the multi-voxel signals that the RSA targeted alleviated such concern. This is because, in the multi-dimensional space, this means that the neural activities patterns for cues that belong to the same category are clustered together, while the responses for cues from different categories are far away from each other. That is to say, unlike the univariate analysis in which one could only look at contrasts which are intrinsically specific to certain categories such as one category > other categories, in RSA all categories in consideration are treated equally by the category model. This significantly reduces the possibility that the category representations we found (Figure 5B) were specific to the categories we actually used. Together with the fact that such category representations overlapped with valuation areas, we feel that these are worth reporting to the research community, with all the caveats that the reviewer has nicely discussed.

Some other minor suggestions for revision.

- In Table 1 - an activation is labelled as “middle frontal gyrus/lateral orbitofrontal cortex”
- these regions are distant from one another and the supplied Talairach coordinate seems to fall within OFC, quite ventral to MFG. Could the authors clarify the reason for labelling this region as MFG?

We thank the reviewer for catching this. This was a mistake – the activation should indeed be OFC, but not MFG. We have corrected Table 1.

- In Figs 1-3: Y-axes declare that Z-coefficients are presented, but if they are Z-scores, then values are quite low (0.1-0.4 in many cases) for the significance levels given. Is it possible that these are actually raw R-values prior to transformation to Z-values? Could the authors please clarify?

We apologize for the confusion. What we plotted in Figures 2-4 are mean beta values (regression coefficients/parameter estimates averaging across participants) from the GLMs, not raw z-scores for the statistical tests we conducted. The units of the coefficients (parameter estimates) the beta values are on the same scale as z-scores because the BOLD data was z-transformed before fitting the GLM. These beta values were then subject to either regressions against pleasantness ratings (Figures 2B & 3B) or to t-tests against zero (Figures 2C, 3C, & 4). Therefore, the numerical values of these coefficients are not related to the significance levels. To clarify this, we have revised the axis labels and added the explanation above to the Methods section (page 25):

“For reporting results from the ROI analyses, mean beta values (regression coefficients/parameter estimates averaging across participants) from the GLMs are plotted (Figures 2B/C, 3B/C, and 4, Supplementary Figures 2 and 4). The units of the beta values are on the same scale as z-scores because the BOLD data was z-transformed before fitting the GLM. These beta values were then subject to subsequent second-level statistical tests, such as regressions against variables of interest, paired t-tests between conditions, or t-tests against zero.”

1. Rice GE, Lambon Ralph MA, Hoffman P (2015): The Roles of Left Versus Right Anterior Temporal Lobes in Conceptual Knowledge: An ALE Meta-analysis of 97 Functional Neuroimaging Studies. *Cerebral cortex*. 25:4374-4391

REVIEWERS' COMMENTS:

Reviewer #1 (Remarks to the Author):

The responses to reviewers are marginal.

1. Context/Categories - The use of the example is only hindering the reader's understanding (not misrepresenting the results) and it is possible I am misinterpreting but there is still a confusing mismatch between the author's examples, results descriptions, and their operationalization. Ice cream (reward item belonging to the category food) may be more valuable on a hot day (context) vs cold day (context) but is not the same as talking about comparing ice cream (reward) to praise from a colleague (alternative modality for reward, maybe a separate category). The authors describe an interaction between context and category but manipulate only category but in the experiment effectively manipulate amount of ice cream or type of ice cream. If clarity in the paper matters, change the initial example and check for other language in the results and discussion that slips back into the contextual framework.

2. To norm or not - If there is a mean bias then salience and value are potentially conflated regardless of whether or not normalization is the right procedure. If salience and value cannot be separated because of the bias, then the results should be described with that caveat. The alternative would be to demonstrate the results are consistent using normed data (then the results are consistent independent of assumptions).

4. Great! Glad to see the analysis scripts posted. I am sure it will help others replicate your work.

Reviewer #2 (Remarks to the Author):

The authors have extensively answered all the critiques.

Reviewer #3 (Remarks to the Author):

The authors have made responsive revisions following the feedback from the 3 reviewers. I believe the manuscript is much improved as a result and have no further concerns to present. I would support acceptance of the manuscript in its present, revised form.

1. Context/Categories - The use of the example is only hindering the reader's understanding (not misrepresenting the results) and it is possible I am misinterpreting but there is still a confusing mismatch between the author's examples, results descriptions, and their operationalization. Ice cream (reward item belonging to the category food) may be more valuable on a hot day (context) vs cold day (context) but is not the same as talking about comparing ice cream (reward) to praise from a colleague (alternative modality for reward, maybe a separate category). The authors describe an interaction between context and category but manipulate only category but in the experiment effectively manipulate amount of ice cream or type of ice cream. If clarity in the paper matters, change the initial example and check for other language in the results and discussion that slips back into the contextual framework.

We have revised the example in the Introduction (page 4 of the main text) following the reviewer's suggestion. This part no longer mentions context and its interactions with category, and it now reads as follows:

“Information about the nature and category of each option, however, is integral to making optimal decisions. Category information may be a key input to the computation of value, guided by the current motivational goals. For example, an ice cream and a magazine may both be valuable, but depending on your hunger level, you may prefer one or the other. We therefore hypothesized that category information may coexist with value signals to facilitate their flexible integration.”

2. To norm or not - If there is a mean bias then salience and value are potentially conflated regardless of whether or not normalization is the right procedure. If salience and value cannot be separated because of the bias, then the results should be described with that caveat. The alternative would be to demonstrate the results are consistent using normed data (then the results are consistent independent of assumptions).

We have added a note in the Supplementary Information (Supplementary Note 1 and Supplementary Fig. 1, adapted from our previous response to this comment) to discuss the caveat with our approach for the pre-defined neutral point. We now point the readers to Supplementary Note 1 when describing the saliency results in the main text (page 6). Supplementary Note 1 and Supplementary Fig. 1 are also pasted below for convenience:

“Supplementary Note 1: Calculation of saliency

Our main analysis assumed the same predefined neutral rating (5) for all participants. A potential concern is that different participants may use different neutral points in their value estimations. Here we elaborate on the rationale for this choice and contrast it with alternative options, such as mean centering and normalization.

First, as explained in the main text, when we introduced the rating scale to the participants during the pre-scan instructions, we did provide a standard interpretation of the ratings by emphasizing that 5 was neutral, and anything above/below 5 would be pleasant/unpleasant (these interpretations also appeared on the screen next to the scale in every trial). Admittedly, there is no objective way of verifying that all subjects used the scale in the way we asked them to, but it seems appropriately conservative to assume so without further information of a systematic bias. Second, in both mean centering and normalization, using

the mean rating as the neutral point also implicitly assumes that overall the positive and negative outcomes in our paradigm cancelled each other out in terms of deviations from neutrality, which may not necessarily be true. To illustrate this, let us imagine a participant who uses the rating scale with 5 as her subjective neutral point, and gives monetary gains, monetary losses, and pleasant faces average ratings of 7, 3, and 7, respectively. Meanwhile, this participant is particularly sensitive to pain, so the electric shocks are more negative (and more salient) to her. As a result, her average rating for electric shocks is 1.5 (still with 5 being neutral). Under such circumstance, the mean rating across all categories for this subject will be below 5, but using this mean as the neutral point to calculate saliency would introduce much bias – the saliency for the two negative categories would be lower, while the two positive categories would have inflated saliency.

Supplementary Figure 1. (Related to Figures 1D and 3) **Individual-level distribution characteristics of pleasantness ratings**

A. Histogram of individual-level mean cue ratings across all categories.

B. Histogram of individual-level standard deviation of ratings across all categories.

With such caveat in mind, we can also examine the distribution of subject-level mean ratings in our behavioral data, to search for participants (if any) who only reported very high or very low ratings (Supplementary Figure 1A). It can be seen that the subject-level mean ratings were indeed distributed within a relatively narrow range around 5, and the ‘mean of the mean’ was 4.93. This indicates that our dataset did not have extreme cases where only one end of the scale was used (e.g. a ‘pessimistic’ subject using 5 as the highest response).

In addition to mean centering, normalization also involves rescaling centered data by the standard deviation, which essentially means adjusting the saliency by the range of ratings of individual participants in this context. The distribution of subject-level standard deviations of ratings is presented in Supplementary Figure 1B, where a moderate degree of individual differences can be observed. It is important to recognize the two potential sources of such differences: individual differences in the usage of the scale and/or individual differences in stimuli/outcome evaluation. For example, two subjects, A and B, had ratings ranging from 1 to 9 and 3 to 7, respectively. While it is possible that the two of them had the same experience in the task and only differed in how they used the rating scale, it is also possible that subject A did have a more intense affective experience (thereby having a greater dynamic range for ratings) than subject B. In the latter case, assuming that a rating of 9 for A indicates the same level of pleasantness and salience as a rating of 7 for B would be detrimental for examining the neural representation of these quantities across subjects.

To summarize, calculating saliency by mean centering or normalization (z-scoring) entails making a set of strong assumptions of how ratings map onto subjective value/saliency, which are hard to verify. More broadly speaking, these issues are general limitations of working with ratings. We believe that our approach is a more conservative choice given how the experiment was designed and administered, as well as the participants' behavior. It is important that readers evaluate our results with these caveats in mind."